# ViTaS: Visual Tactile Soft Fusion Contrastive Learning for Reinforcement Learning

## Abstract

Tactile information plays a crucial role in human manipulation tasks and has recently garnered increasing attention in robotic manipulation. However, existing approaches struggle to effectively integrate visual and tactile information, resulting in sub-optimal performance. In this paper, we present **ViTaS**, a simple yet effective framework that incorporates both visual and tactile information to guide an agent's behavior. We introduce *Soft Fusion Contrastive Learning*, an advanced version of conventional contrastive learning method, to enhance the fusion of these two modalities, and adopt a CVAE module to utilize complementary information within visuo-tactile representations. We conduct comprehensive experiments, including **9** tasks in simulation environment, across **5** different benchmarks, to compare ViTaS with existing baselines. The results demonstrate that ViTaS achieves state-of-the-art performance, with an average improvement of **51**%. Furthermore, our method significantly enhances sample efficiency while maintaining minimal parameters, underscoring the effectiveness of our approach. The code will be released upon acceptance.

## 1 Introduction

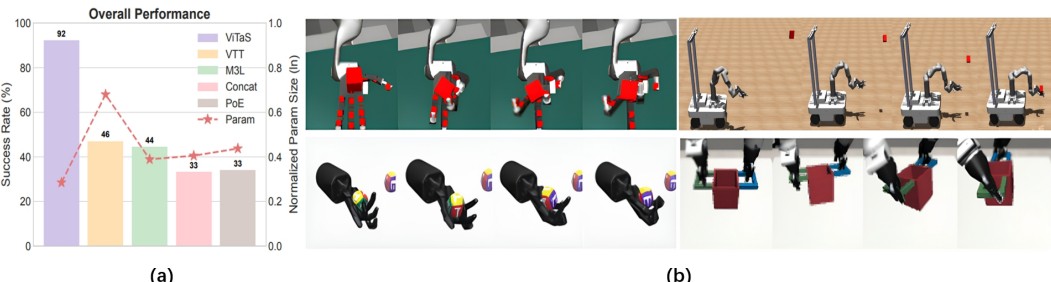

Figure 1: **Overall performance.** (a) shows the overall average success rate and feature extractors parameter size (in a log scale). (b) shows success demos for 4 tasks (Block Spin, Mobile Catch, Egg Rotate, Dual Arm Lift).

Humans are adept at performing complex manipulation tasks, such as spinning a pen or lifting a block. While vision plays a critical role in these activities, other modalities, particularly touch, also provide rich information for object manipulation. Interestingly, visual and tactile information often exhibit significant relevance and complementarity (Apkarian-Stielau & Loomis, 1975). For individuals with visual impairments, a clearer mental reconstruction of an original visual image can be achieved by combining a blurred visual perception with tactile information (Kappers, 2011).

Most previous reinforcement learning (RL) algorithms have relied primarily on visual information to address manipulation tasks (Yuan et al., 2022b; Xiao et al., 2022; Yuan et al., 2022a; Li et al., 2024b;c; Haarnoja et al., 2024; Yu et al., 2023; Pitz et al., 2023; Lin et al., 2024; Xu et al., 2023; Qi et al., 2023b;a; Plappert et al., 2018b; Yuan et al., 2024b). Recently, several efforts have aimed to incorporate tactile information to improve the performance of RL algorithms. However, these approaches generally exhibit limited fusion between the two modalities. For instance, Sferrazza et al.

(2023) directly concatenates visual and tactile inputs and feeds them into MAE, while Chen et al. (2022) segments visual and tactile data into patches and uses a transformer to extract representations. As a result, these methods often demonstrate limited performance in contact-rich manipulation tasks that rely heavily on both visual and tactile inputs, such as in-hand rotation. Moreover, many previous methods employ complex encoders like transformers and MAE, which involve intricate architectures with numerous parameters, leading to prolonged training time. Due to the insufficient utilization of tactile information, these methods also diminish sample efficiency of RL algorithms in guiding manipulation. Given these limitations, we pose the question: *how can we more effectively fuse visual and tactile information, to enhance the performance of RL algorithms for manipulation*?

Drawing on prior research in human physiology regarding the processing of visuo-tactile information, we propose **Vi**sual **Ta**ctile **S**oft Fusion Contrastive Learning (ViTaS), a novel visuo-tactile representation learning framework for reinforcement learning. Generally ViTaS can be divided into two parts. Firstly, given the inherent relevance between visual and tactile modalities, we utilize contrastive learning to align the embeddings of visual data with their corresponding tactile information in the latent space. Notably, we employ *soft fusion contrastive learning* inspired by Han et al. (2020) to fuse features in alternated modalities. Specifically, we extend the original RGB single-modality framework to incorporate both visual and tactile modalities, enabling the agent to leverage samples of different timesteps with similar tactile information as positive samples. Additionally, inspired by the ability of humans to reconstruct clear images from blurred visual inputs combined with tactile information and complementarity of two modalities, we integrate conditional variational autoencoder (CVAE) introduced by Sohn et al. (2015) to reconstruct the original image with the embeddings of vision and touch, further improving the fusion of visual and tactile information.

To evaluate the performance of our algorithm, we conduct experiments on **9** tasks across **5** environments: Insertion (Sferrazza et al., 2023), Gymnasium (Towers et al., 2024), Robosuite (Zhu et al., 2020), Mobile Catch (Zhang et al., 2024) and Block Spin (Yuan et al., 2024a). Especially, the original Mobile Catch and Block Spin environments lack tactile information, so we integrate tactile sensors into the robotic hand, creating 2 challenging visuo-tactile tasks. Additionally, to demonstrate the generalization capability of our system, we perform further experiments on 3 auxiliary tasks, as well as several ablation studies. The overall experimental results illustrated in Figure 1 and Figure 2 (a) show that ViTaS achieves state-of-the-art performance compared to other visuo-tactile learning methods in all tasks, with an average success rate of 92% and average improvement of 51%, while minimizing total trainable parameters in representation learning.

In summary, our contributions are as follows:

- We improve the traditional contrastive learning method and leverage it for the fusion of visual and tactile modalities.
- We propose ViTaS, a simple yet effective representation learning paradigm that can integrate visual and tactile inputs through soft fusion contrastive as well as CVAE, and utilize it to guide the training of reinforcement learning and visuomotor control agent.
- We evaluate our algorithm on various simulation tasks, demonstrating state-of-the-art performance with an average growth rate of 51%. ViTaS significantly improves sample efficiency and also minimizes the number of parameters in the feature extractor.

## 2 RELATED WORK

**Visuo-Tactile Representation Learning** In recent years, numerous cross-modal representation learning methods, particularly those focused on visuo-tactile integration, have emerged, as demonstrated by Lee et al. (2020; 2019); Dave et al. (2024); Yang et al. (2024); Polic et al. (2019); Xu et al. (2024); Li et al. (2022a); Yang et al. (2023); Lin et al. (2024); Xu et al. (2023); Radford et al. (2021); Li et al. (2024a; 2022b). Among them, Li et al. (2019) utilizes an adversarial loss to learn representation in the latent space, while Chen et al. (2022) leverages a transformer architecture to integrate multiple modalities, introducing alignment and contact loss to enhance performance. Sferrazza et al. (2023) proposes a jointly visuo-tactile training scheme using an MAE-based encoder trained through a reconstruction process, with the encoder co-trained for policy learning.

Despite the success of these approaches in specific tasks, they often fail to fully exploit the correspondence between visual and tactile modalities, leading to sub-optimal encoder training and re-

duced success rates in tasks such as dexterous hand manipulation. In contrast, our method employs a simpler yet highly effective CNN-based encoder to improve the alignment and fusion of modalities, achieving superior performance across multiple benchmark tasks.

**Contrastive Learning** Extended into computer vision by the MoCo series (He et al., 2020; Chen et al., 2021), and SimCLR (Chen et al., 2020), contrastive learning has emerged as a prominent technique for representation learning. We intend to extend the contrastive learning paradigm to visuo-tactile framework for reinforcement learning. Related examples include Wang & Hu (2023a), Dave et al. (2024), Yuan et al. (2021), Laskin et al. (2020), Wang & Hu (2023b), Han et al. (2020), Yang et al. (2024), Wang & Hu (2023a), Yuan et al. (2024a) and Zhan et al. (2022). Among the works most closely related to ours, Dave et al. (2024) proposes a visuo-tactile fusion approach based on contrastive pre-training. Zhan et al. (2022) employs contrastive loss within the visual modality to enhance policy learning. Yang et al. (2024) incorporates tactile, vision and text using contrastive learning to solve downstream tasks. Han et al. (2020) advances contrastive learning paradigm utilizing top $K$ analogous samples in optical flow as positive samples in RGB modality.

However, as Han et al. (2020) mentions, simply doing instance discrimination tends to neglect some key information that two resembling samples may be negatives for each other due to distinct timesteps. The phenomenon also pops up in the field of cross-modal contrastive learning. We refine the contrastive learning method to alleviate the issue, which is elaborated in Section 3.1.

# 3 METHOD

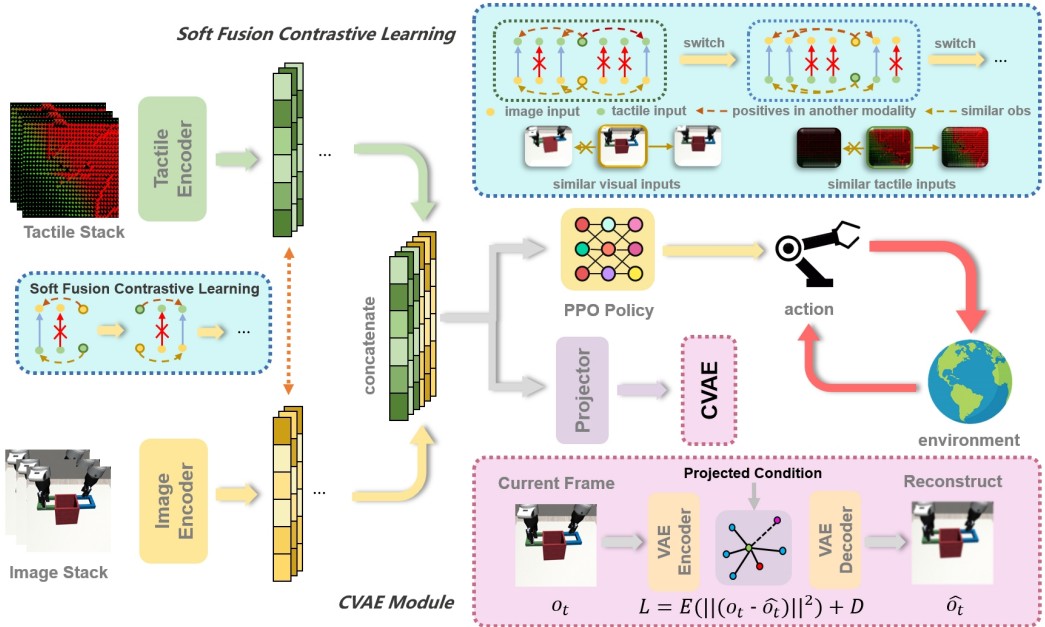

Figure 2: **Method overview.** The agent takes information from two modalities, visual and tactile, as inputs, which are then processed through separate CNN encoders. Encoded embeddings are utilized by cross-modal soft fusion contrastive approach, yielding fused feature representation for policy network. A CVAE-based reconstruction framework is also applied for cross-modal integration.

In this section, we elaborate **Vi**sual **Ta**ctile **S**oft Fusion Contrastive Learning (ViTaS), an advanced visuo-tactile fusion framework tailored for reinforcement learning. We note that when two tactile maps are resembling, the key features derived from the corresponding visual data should likewise bear a strong resemblance and vice versa. This alignment makes the data particularly well-suited for contrastive learning. Meanwhile, given the complementary information tactile and image offer, our objective is to reconstruct the features extracted from two modalities, obtaining outputs with enriched and more detailed information. CVAE is a good candidate for this process. Therefore, ViTaS

fuses visual and tactile modalities through the collaboration of Cross-modal Soft Fusion Contrastive Learning (Section 3.1) and CVAE (Section 3.2). We utilize the PPO (Schulman et al., 2017), an on-policy reinforcement learning strategy, for the underlying algorithm framework of our method. Formally, our ultimate reinforcement learning objective can be defined as follows:

$$\mathcal{L} = \lambda \mathcal{L}_{\text{CON}} + \mu \mathcal{L}_{\text{VAE}} + \mathcal{L}_{\text{PPO}}, \tag{1}$$

where $\lambda, \mu$ be tunable parameters to bridge the gap between various components. The value of $\mu$ is well studied in Appendix A.10. Critical hyper-parameters and detailed settings of CNN are displayed in Table 3 and Table 4. The overall pseudocode for ViTaS is shown in Algorithm 3.

### 3.1 SOFT FUSION CONTRASTIVE LEARNING

We denote a trajectory as $\Gamma = \{o_i, t_i\}_{i=1}^N$ where $o_i$ stands for image observation at $i$-th timestep and $t_i$ for tactile inputs, with total length $N$. For simplicity, we denote $o_i$ and $t_i$ are *dual* samples of each other. We use two convolutional neural networks separately to extract features from raw images and tactile maps. Formally, we denote $f_o(\cdot)$ and $f_t(\cdot)$ as the image and tactile extractors respectively.

Inspired by Han et al. (2020), we present *soft fusion contrastive learning*, a novel cross-modal contrastive learning paradigm to enhance the fusion of two modalities. We use *soft fusion contrastive* below for simplicity. Specifically, we accomplish this by identifying the $K$ most analogous samples from one modality, say modality $\mathcal{A}$, leveraging their *dual* samples as positives for each other in alternated modality $\mathcal{B}$. During the process, the parameters of the encoder corresponding to modality $\mathcal{A}$ are frozen, with the counterpart in $\mathcal{B}$ updated actively. $K$ is a hyper-parameter representing the number of positives needed to be utilized. We set $K = 10$ by default and the effectiveness of different values of $K$ will be studied in Section 4.4. Then, we reach the following formula in accordance to the description above:

$$\begin{cases} \mathcal{L}_{\text{CON,1,i}} = -\mathbb{E}\left[\log \dfrac{\sum_{p \in \mathcal{P}_1(i)} \exp(f_o(o_p) \cdot f_o(o_i) \, / \, \tau)}{\sum_{p \in \mathcal{P}_1(i)} \exp(f_o(o_p) \cdot f_o(o_i) \, / \, \tau) + \sum_{n \in \mathcal{N}_1(i)} \exp(f_o(o_n) \cdot f_o(o_i) \, / \, \tau)}\right] \\ \text{s.t. } \mathcal{P}_1(i) = \{j | (\text{Sim}(f_t(t_j), f_t(t_i))) \in topKmax_k(\text{Sim}(f_t(t_i), f_t(t_k)))\}, \ \mathcal{N}_1(i) = S \setminus \mathcal{P}_1(i) \end{cases} \tag{2}$$

In the formula, we denote $\mathcal{P}_1(i)$ as the set of positives of $o_i$, calculated by $K$ most similar samples in corresponding inputs in tactile modality, while $\mathcal{N}_1(i)$ as negatives. $S$ stands for universal samples of $o_i$ in one trajectory. We use $topKmax_k(U)$ to obtain the top $K$ similar samples in set $U$, which is obtained in replay buffer in implementation. $\text{Sim}(x, y)$ calculate the similarity between key features $x$ and $y$. We use *cosine similarity* to achieve this, with detailed information elaborating in Appendix A.7. We would like to emphasize that we discern positives by extracted features, where the encoders walk in.

Similarly, we switch the position of $t_i$ and $o_i$ periodically like workflow presented in Yang et al. (2023) to update both encoders equally, and the corresponding metrics are then denoted as $\mathcal{L}_{\text{CON,2,i}}$ and $\mathcal{P}_2(i)$. Moreover, we replace all $o_i, f_o(\cdot)$ in the formula above as $t_i, f_t(\cdot)$ and vice versa.

To achieve a more balanced update of the target, we adopt alternating updates when calculating ultimate objective $\mathcal{L}_{\text{CON}}$ according to $\mathcal{L}_{\text{CON,1/2,i}}$. Specifically, $\mathcal{L}_{\text{CON}}$ is contributed by $\mathcal{L}_{\text{CON,1,i}}$ at start, and shifted to $\mathcal{L}_{\text{CON,2,i}}$ after exact $T_{\text{switch}}$ steps and so forth. Furthermore, we define the coefficient sequence as $u_i = 1/2 \times \left(1 - (-1)^{\lceil i/T \rceil}\right) = [1, 1, \cdots, 1, 0, 0, \cdots, 0, 1, 1, \cdots]$. Consequently, the target of the contrastive loss can be written as:

$$\mathcal{L}_{\text{CON}} = \sum_{i=1}^N u_i \cdot \mathcal{L}_{\text{CON,1,i}} + (1 - u_i) \cdot \mathcal{L}_{\text{CON,2,i}} \tag{3}$$

To facilitate a more thorough and specific comprehension of the part, we have included the pseudocode of soft fusion contrastive in Algorithm 1.

### 3.2 Conditional VAE Visuo-Tactile Feature Integration

In the realm of visuo-tactile integration, VAE-based methods are commonly employed (Nair et al., 2018; Bai et al., 2021). Inspired by Bachhav et al. (2019), we extend the CVAE framework for visuo-tactile fusion by incorporating the *condition* component, which is derived from the projection of image and tactile embedding. Consequently, the image and tactile encoders are optimized concurrently during the training process. A comprehensive depiction is presented in Figure 2.

We establish *condition* on the concatenated visuo-tactile feature $c$ to reconstruct the current image frame $o_{\text{cur}}$. CVAE consists of an encoder $p_\theta(\cdot)$, decoder $q_\psi(\cdot)$, and visuo-tactile embedding projector $f_\phi(\cdot)$, which are parameterized by $\theta$, $\psi$ and $\phi$ separately. We use $z$ to represent the latent variables, and the reconstructed frame $\hat{o}_{\text{cur}}$ conditioned on visuo-tactile feature $c$ can be expressed as:

$$\hat{\mathbf{o}}_{\text{cur}} = q_\psi(p_\theta(o_{\text{cur}}, f_\phi(c)), f_\phi(c)) \tag{4}$$

In accordance with CVAE constraints, the target can be formulated as:

$$\mathcal{L}_{\text{VAE}} = \mathbb{E}\left[\|o_{\text{cur}} - \hat{o}_{\text{cur}}\|^2\right] + D_{\text{KL}}\left(p_\theta(z|p_\theta(o_{\text{cur}}), c)\| \mathcal{N}(0, 1)\right) \tag{5}$$

Notably, the CVAE module is active only during training and does not impose any additional computational overhead during test time.

## 4 Experiments

In this section, we conduct several tasks, most of which are relatively complex and contact-rich, to showcase the performance in cross-modal fusion and policy learning. We aim at clarifying following questions through comprehensive experiments:

  (i) Does ViTaS have the capability to solve complicate tasks requiring compact tactile information (e.g. dexterous hand rotation)?

 (ii) How does ViTaS demonstrate generalization and robustness with respect to tasks involving objects of various shapes, significant noise or different physical parameters?

(iii) What merit does ViTaS offer in comparison to previous visuo-tactile algorithms?

Moreover, we do a thorough examination of distinct components contributing to overall performance of ViTaS, and the detailed analysis is presented in Ablation Study (see Section 4.4).

### 4.1 Simulation Environment

We conduct experiments using 9 simulated tasks, categorized into 5 primary parts and shown in Figure 4: (a). shadow dexterous hand tasks (Plappert et al., 2018a; Melnik et al., 2021) based on Gymnasium (Towers et al., 2024) (pen rotation, block rotation, and egg rotation), (b). Robosuite (Zhu et al., 2020)-based tasks (door opening, lift, and dual arm lift), (c). Insertion tasks originated by Sferrazza et al. (2023) simulated in mujoco, (d). Mobile-Catch environment implemented by Zhang et al. (2024) and (e). Block Spinning task created by Yuan et al. (2024a). Beyond these foundational experiments, we introduce a series of auxiliary tasks involving altering object shapes in Lift or modifying physical parameters in Pen Rotation. The outcomes of (a)-(d) environments are quanti-

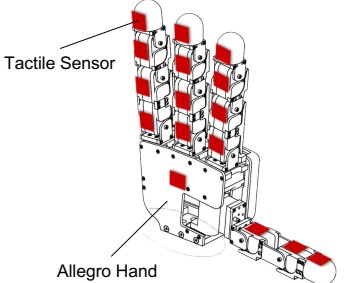

Figure 3: **Tactile Sensors on Allegro Robotic Hand.**

fied in terms of success rate, whereas (e) is assessed based on training reward. A comprehensive explanation of the experimental settings is provided in Appendix A.1.

Moreover, it is crucial to integrate tactile sensors to obtain tactile data for ViTaS framework. In recent years, tactile sensors have gradually piqued the interest of researchers. For real-world scenarios, sensors like Sferrazza & D'Andrea (2022), Zhang et al. (2022), Lin et al. (2023b), Lin et al. (2023a) are commonly used, whereas in simulation environments, Akinola et al. (2024), Si & Yuan (2022) are utilized. In our experiments, we incorporate appropriate tactile sensors across all 9 environments.

Egg Rotate    Block Rotate    Pen Rotate    Door    Dual Arm Lift    Lift    Insertion    Mobile Catch    Block Spin

(a)                              (b)                              (c)    (d)    (e)

Figure 4: **Tasks.** Our method is evaluated on 9 contact-rich tasks across 5 different domains, covering multiple types of grippers and dexterous hands, with tactile sensors embedded.

For the 3 Gymnasium rotate in-hand rotation tasks, we employ the built-in tactile modules. For the Lift, Insertion, and Door Opening tasks, we employ a parallel gripper equipped with a $32 \times 32 \times 3$ tactile sensor at the contact surfaces between the gripper and the object. Among the 3 channels in tactile map, channel 1 and 2 represent the normal force and the value of channel 3 denotes shear force. The kind of tactile sensors is adopted following that in Taylor et al. (2022) and Xu et al. (2023). In the catch and block spin tasks, we enhanced the Allegro hand and Leap hand with tactile sensors by integrating four $3 \times 3 \times 3$ sensors on each finger (located at the proximal, middle, distal, and tip segments) and one $3 \times 3 \times 3$ sensor on the palm, as shown in Figure 3. These sensors are zero-padded to form a $32 \times 32 \times 3$ input, following Sferrazza et al. (2023).

### 4.2 COMPARED BASELINES

To validate the effectiveness of ViTaS, we compare with the following visuo-tactile representation learning baselines:

- M3L (Sferrazza et al., 2023): A visuo-tactile fusion training algorithm utilizing the MAE encoder for PPO policy learning, while simultaneously trains the MAE encoder with reconstruction loss through the MAE encoder-decoder architecture. Information in visual and tactile is concatenated and passed through a unified encoder. Training process of policy and MAE encoder occurs concurrently.

- VTT (Chen et al., 2022): A visuo-tactile fusion training method rooted in the transformer architecture. Specifically, both image and tactile data are segmented into patches, which are then processed through transformer layers to obtain embeddings. Subsequently, the latent representations are reconstructed into images and rewards using a VAE decoder. The method also incorporates contact and alignment modules to further enhance performance.

- PoE (Lee et al., 2020): A VAE-like framework to fuse two modality. In particular, subsequent to acquiring the embeddings, instead of reconstructing images via a decoder, an MLP is deployed after the decoder stage to tackle designed tasks. Additionally, KL divergence is harnessed to ascertain the parameters of the latent embeddings.

- Concatenation (Lee et al., 2019): A relatively primitive multi-modal fusion method. Image and tactile data are passed through a CNN encoder independently, whose results are then concatenated and fed into a contrastive-like module.

### 4.3 EXPERIMENT RESULTS

We conduct comparisons of our algorithm against 4 baseline methods across the aforementioned 9 primitive tasks. We evaluate each algorithm in each environment 5 times under different random seeds, and average the results when training $3 \times 10^6$ timesteps to obtain the performance metrics.

The results for all 5 methods in 9 tasks are shown in Figure 5. Several baselines show excellent performance in some relatively simple tasks like Door Opening or Insertion. However, for tough tasks like Egg Rotation and Block Rotation, which are contact-rich and require methods to incorporate visual and tactile information jointly, few baselines can solve it within a limited horizon, while ViTaS still owns the capability to succeed. We can calculate the average success rate for ViTaS and other baselines in Figure 5, obtaining the results in Figure 1. The average improvement upon 4 baselines is 51%, also illuminated in the abstract part. Furthermore, in cases like the Dual Arm Lift, where all baselines are capable of solving the task, ViTaS exhibits notably enhanced sample efficiency by

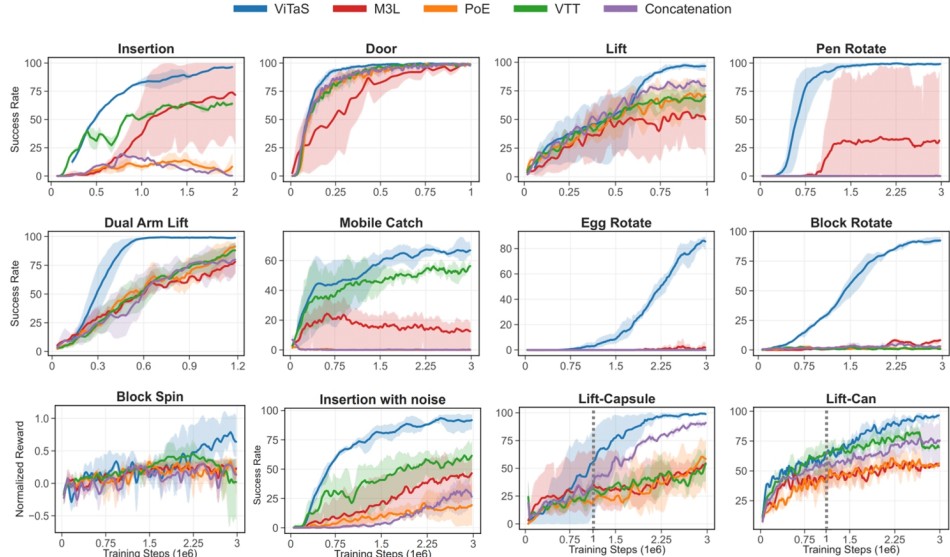

Figure 5: **Learning curves for all** 5 **methods in** 9 **primitive tasks and the derived ones.** We utilize success rate for the evaluation of first 8 tasks, with reward for Block Spin task. In the last 3 plots, we measure model robustness on additional Gaussian noise in Insertion and various objects in Lift.

accomplishing the desired outcomes in far fewer timesteps (about 3 times faster). **This underscores its exceptional capability to extract features and solve complicate tasks, clarifying question (i)**.

In order to assess the generalization capability and robustness of our approach, we introduce auxiliary tasks derived from the Lift and Pen Rotate tasks mentioned earlier. For the Lift task, the object shape is modified from a cube to cylinder and capsule in both training and testing phases, allowing us to evaluate the method's resilience to changes in object geometry. As for the Pen Rotate task, we randomize the target angle within a large range, enabling a thorough evaluation of the model's generalization across varying conditions. We also add Gaussian noise with 0.3 standard deviation in Insertion task (The intensity of Gaussian noise of different standard deviation could refer to Figure 12. The experimental parameters are kept in alignment with the preceding 9 primitive benchmarks.

As illustrated in Figure 5, when the object shape is changed, every baseline model experiences a performance drop when training $10^6$ timesteps, indicating sensitivity to these alterations. ViTaS, however, exhibits a negligible decrease, demonstrating its resilience to variations in object geometry. Pen rotation, among the most challenging tasks, is only successfully handled by ViTaS and M3L. When the target angle is randomized, the agent is required to extract the most significant information from the current observations via visual tactile representation learning to make appropriate moves. In this scenario, M3L struggles to maintain per-

Table 1: **Generalization ability.** We show the success rate of best run among 3 seeds in Pen Rotate task with randomized position. All results are recorded when training $3 \times 10^6$ timesteps.

| Tasks / Method | ViTaS | M3L |
|---|---|---|
| Fixed | **99.2** | 93.1 |
| Random | **78.4** | 62.7 |
| Drop | **20.8** | 30.4 |

formance, while ViTaS continues to solve the task effectively with less drop than M3L. Given the robust performance in 3 auxiliary tasks, **we provide a clear clarification of question (ii)**.

Another observation from Figure 5 is that although baseline like M3L can solve tasks in some cases, it may suffer from unstable training process and large variance (e.g. insertion, door and pen rotate). A possible explanation lies in M3L's approach of segmenting both image and tactile maps into discrete tokens, which are then passed through the MAE encoder. This token-based feature extraction is prone to overlooking critical information compared to pixel-level extraction, thus resulting in the capture of fewer essential features. The dexterous hand manipulation task, however, requires more compact information especially rich tactile features in latent space to control

the manipulation agent, and the rather scarce information of observation leads to underfitting of the agent and large variance of training metrics.

We quantify the number of trainable parameters employed in representation learning across 5 algorithm, depicted in Figure 1. It is evident that ViTaS, in addition to exhibiting superior performance, also owns a smaller parameter count compared to other baselines.

Taking into account several key reasons including higher success rate and sample efficiency, fewer training parameters, enhanced generalization and robustness and diminished training variance, **we reach the conclusion that ViTaS owns better performance in all presented benchmarks, showing a state-of-the-art result, which is a comprehensive solution to question (iii).**

## 4.4 ABLATION STUDY

To verify the fidelity of various designs in our algorithm, we conduct extensive ablation experiments to show the necessity of each component. The overall ablation results are presented in Table 2. In order to simplify the name of each ablation experiments, we use abbreviations of experiments in the first row of Table 2. Specifically, the corresponding experiments are ViTaS, w/o. **Ta**ctile, w/ **U**nified Encoder, w/o. soft fusion **C**ontrastive module, w/ **T**ime **C**ontrastive, $K = 1$ and $K = 50$. The meaning and detailed information of each experiment are clarified in the following sections.

Table 2: **Overall ablation study.** We conduct the experiments with each repeats with 5 times, and take the average success rate percentage as results when training $3 \times 10^6$ times. The experiment results show that various designs in ViTaS are of high importance. Without the designs, the success rate drops **31**.5% on average. Names of each column are abbreviations of different experiments, and are explained above. Green for optimal results while purple for suboptimal.

| Tasks / Methods | V | w/o. TA | U | w/o. C | TC | K1 | K50 |
|---|---|---|---|---|---|---|---|
| Insertion | $99.2_{\pm 0.7}$ | $88.1_{\pm 4.3}$ | $61.6_{\pm 5.7}$ | $90.3_{\pm 2.5}$ | $75.2_{\pm 5.2}$ | $83.3_{\pm 3.9}$ | $78.7_{\pm 3.6}$ |
| Block rotation | $92.7_{\pm 2.0}$ | $67.7_{\pm 3.6}$ | $18.4_{\pm 0.8}$ | $67.7_{\pm 2.6}$ | $79.5_{\pm 4.3}$ | $88.0_{\pm 2.1}$ | $70.1_{\pm 4.8}$ |
| Egg rotation | $85.3_{\pm 3.1}$ | $24.3_{\pm 4.1}$ | $3.3_{\pm 2.4}$ | $6.5_{\pm 2.1}$ | $57.7_{\pm 1.8}$ | $65.2_{\pm 2.3}$ | $3.6_{\pm 6.4}$ |
| Average | $92.5_{\pm 2.8}$ | $60.9_{\pm 5.6}$ | $27.1_{\pm 2.6}$ | $54.7_{\pm 6.1}$ | $70.6_{\pm 4.4}$ | $78.8_{\pm 3.7}$ | $67.3_{\pm 5.2}$ |

**Is tactile information crucial?** We conduct 2 main experiments in this part. Firstly we eliminate the tactile information, retaining only the visual data, and solely utilize the image encoder, while handling the corresponding tactile information through zero-padding. Additionally, the workflow outlined in Sferrazza et al. (2023) employs a unified MAE encoder across both modalities, disregarding the inherent distinctions between them. This oversight could potentially leads to less discriminative feature representations and a notable reduction in overall effectiveness. To prove that tactile maps can offer unique information beyond visual inputs can provide, we build another experiment that image and tactile are directly concatenated and subsequently fed into a shared encoder.

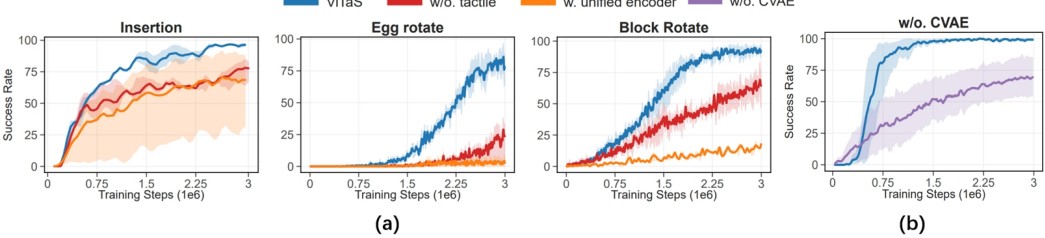

Figure 6: **Learning curve for tactile and CVAE.** The left part shows results for ViTaS, ViTaS w/o. tactile information and w/ unified encoder, while the right one indicates pen rotation ablating CVAE.

The results in Figure 6 (a) show that when ablating tactile information, the success rate in 3 benchmarks drops 34% on average. Thus, tactile information gets crucial in dexterous operation tasks like rotation, while it also makes difference in simpler tasks like Insertion.

Using a unified encoder, however, is not a good choice either given the poor performance on the U-column in Table 2, almost failing to accomplish 2 in-hand rotation tasks. We then clarify that tactile has some complementary information to image, which cannot be extracted via a unified encoder.

**How much do CVAE and soft fusion contrastive contribute to ViTaS?** In order to clarify the effectiveness of each component, we remove the CVAE and soft fusion contrastive components separately, conducting independent tests on the same benchmarks and comparing results.

We use ViTaS without CVAE to perform pen rotation task and results are shown in Figure 6 (b). Several extra experiments are also added in Figure 11 and Appendix A.6. The learning curves show that when ablating CVAE, the performance drops heavily (about 25%), and the training process is rather unstable. Moreover, as shown in Table 2, we remove soft fusion contrastive learning and the results drop for about 28.9%, with a surge in variance. We also study the impact of difference value of coefficient in CVAE loss presented in Equation (1), which is shown in Appendix A.10.

Moreover, feature space distribution is shown in Figure 7 to illuminate underlying reasons. We extract visuo-tactile features using ViTaS and ViTaS ablating soft fusion contrastive learning from two aligned trajectories with different plug-in shapes. In the figure, each color represents a trajectory-method combination, with shades from light to dark indicating the progression from start to finish. The endpoints of the lines denote the corresponding visual and tactile embeddings. ViTaS maintains a consistent structure across both trajectories, with similar visuo-tactile embedding relationships and spatial distributions before and after changing the plug-in shape. In contrast, the model without soft fusion contrastive learning lacks this stable relationship, highlighting the superiority of this module in extracting high-quality, generalizable features. By combining the three experiments above, we show that two main components of ViTaS are necessary and powerful.

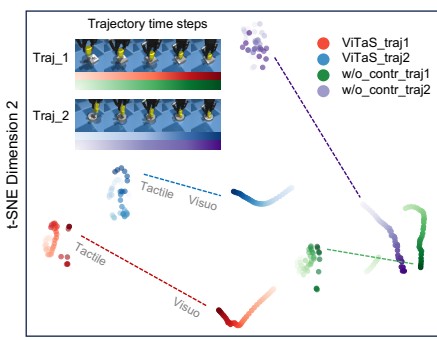

Figure 7: **Visuo-Tactile embedding visualization.** We collect 2 aligned insertion trajectories with different hole shapes. ViTaS exhibits superior capability in extracting structured multimodal features.

$K$ **in soft fusion contrastive learning**. We explore the impact of varying $K$, for instance, setting it to 1, 10 (ours) and 50, to observe how the results are affected. It is noteworthy that image and tactile at the same timestep are the only positives for each other when $K = 1$, adopting the same process as conventional cross-modal contrastive learning. Therefore, by comparing results between ours and $K = 1$, we can also clarify whether soft fusion contrastive could outperform conventional contrastive learning method.

The last two columns of Table 2 show the effectiveness of different $K$ in ViTaS. The results when $K = 1$ show that though conventional contrastive learning can achieve relatively excellent performance, it still has performance gap with our method (i.e. $K = 10$), while too large $K$ value as 50 also causes a drop in performance.

**Soft fusion contrastive v.s. time contrastive**. To verify the effectiveness of soft contrastive in another perspective, we carry out experiments utilizing an alternative contrastive approach, namely *time contrastive*, to highlight the indispensable role of cross-modal soft fusion contrastive learning. Neighboring frames (i.e., a fixed number of preceding and succeeding frames) are treated as positives in this method, while distant frames serve as negatives, echoing with Sermanet et al. (2018). The im-

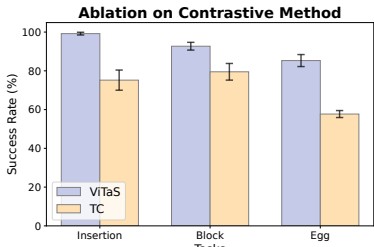

Figure 8: **Ablation on contrastive method.** We evaluate soft fusion contrastive learning (ViTaS) and time contrastive learning (TC) on 3 tasks with 3 seeds when training $3 \times 10^6$ timesteps.

plementation of time contrastive is shown in Algorithm 2 as pseudocode. The motivation behind this lies in emphasizing that, despite frames within close time intervals often appearing to be similar, it is crucial during the contrastive learning process to identify the $K$ most analogous frames, which

may not necessarily be temporally adjacent. This distinction underscores the importance of going beyond mere time contrastive.

As shown in Figure 8, time contrastive learning cannot surpass soft fusion contrastive, proving the necessity of our advanced soft fusion contrastive.

In conclusion, our ablation study delves deep into our algorithm to analyze the effectiveness of each component. The results prove that tactile information, soft fusion contrastive learning and CVAE are of high importance, while soft fusion contrastive performs better than other contrastive methods like conventional contrastive and time contrastive. We show the necessity of every design we use.

## 5   CONCLUSION AND LIMITATIONS

In general, we introduce ViTaS, a succinct yet effective visuo-tactile fusion framework. Drawing an analogy to human physiology, we extend the application of visual and tactile perception to the domain of reinforcement learning, yielding remarkable results. We propose *soft fusion contrastive learning* to extract key features from one modality according to another, and we adopt a CVAE module to utilize complementary information from different modalities. Experiments, auxiliary tasks and ablation studies are meticulously conducted to verify each component of ViTaS. The results show the effectiveness of ViTaS and necessity of soft fusion contrastive and CVAE.

The main limitation of ViTaS is that addressing scalability to more complex manipulation and long-horizon dexterous tasks still remains challenging. Moreover, despite the success in a wide range of tasks in various simulated environments, the applicability in the real world is absent. However, it has not escaped our notice that ViTaS is theoretically capable of being seamlessly deployed to physical robotic systems, for information we use could be easily obtained in the real world. The results in Figure 12 also show the robustness to noise, a main hindrance for real-world applicability, further proving the capability of ViTaS. In the future, we will continually explore the potential of ViTaS for more complex manipulation tasks and validate the transfer on real hardware.

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

# A  APPENDIX

## A.1  DETAILS IN SIMULATED ENVIRONMENT

### A.1.1  INSERTION

The insertion environment consists of various kinds of holes and a peg corresponding to one of those shapes. The peg is attached to a Robotiq-85 robotic arm and securely held by the arms two fingers (grippers). During the operation, tactile sensors fixed on both fingers detect tactile information, generating two $3 \times 32 \times 32$ tactile maps, which is stated in Xu et al. (2023). The three channels represent forces in two shear directions and one contact (normal) direction. A third-person perspective camera is positioned in front of the peg to capture RGB information during the insertion process, represented as a $3 \times 64 \times 64$ image. The task is considered successful when the peg is fully inserted into the hole with the matching shape.

To more effectively evaluate the algorithm's generalization and robustness performance under varying conditions, the insertion environment includes pegs of different shapes (e.g., rectangular, triangular, trapezoidal, etc.). Additionally, multiple positional configurations are provided, ranging from the center to the periphery of the workspace.

### A.1.2  GYMNASIUM-BASED TASKS

Shadow Dexterous Hand in Gymnasium are equipped with tactile sensors on the fingers offered by the sensor package, providing a foundation for tactile-based manipulation operations. We selected three tasks from the Shadow Dexterous Hand suite as our benchmarks, specifically pen rotation, egg rotation, and block rotation.

The tasks are composed of a dexterous hand which is manipulated by agent and an object varying in different task. The target of each task is to rotate the object and reach the final position provided by task itself. For instance, the target in block rotation is to manipulate an egg to reach the target position, which indicated by colors on different faces. To maintain consistency with the baseline, we adopted the environmental configurations outlined in M3L, which include parameters such as position settings, reward magnitude, object specifications, and tactile information padding. Moreover, we add Pen Rotate task with random target position mentioned in Section 4. The accurate angle position of a pen is described as a quaternion in shadow dexterous hand benchmarks. We denote a quaternion as $\{r_x, r_y, r_w, r_z\}$. Initially we randomly generate $\{r'_x, r'_y, r'_w, r'_z\}$ according to following condition:

$$r'_x, r'_y \in [-0.2, 0.2]; \ r'_w, r'_z \in [0.5, 0.8] \cup [-0.8, -0.5]$$

Then, we normalize the quaternion to $\{r_x, r_y, r_w, r_z\}$ so that we have $r_x^2 + r_y^2 + r_z^2 + r_w^2 = 1$, which aligns with the condition needed in shadow dexterous hand.

### A.1.3  ROBOSUITE-BASED TASKS

We selected the Franka robotic arm and a Robotiq 2F-85 gripper from the Robosuite environment, augmenting the gripper fingers with additional tactile sensors to facilitate the acquisition of both visual and tactile data.

In the Door Opening task, the agent manipulates the robotic arm to open a door equipped with a handle. Similar to the aforementioned tasks, we adopt the same environment configuration as used in M3L, employing a dense reward structure. Additionally, the doors initial position is randomized within a specific range: $x \sim [0.06, 0.10], y \sim [-0.1, 0.1]$ to align with baselines and compare fairly.

The above 5 tasks are used in baselines, and we do not modify any parameters. Additionally, we add 2 original tasks based on Robosuite in order to evaluate the performance of algorithms. The two tasks, Lift and TwoArmLift, will be clarified below.

In the Lift task, the robotic arm attempts to learn a strategy for lifting an object, which could be a cube or potentially other shapes like cuboids or cylinders, to a specified height. To better focus on assessing the fusion of multimodal information and testing the models generalization capabilities, we employ a dense reward structure to facilitate learning. Specifically, when the robotic arm successfully lifts the object, the reward is increased to 300, and when the object is successfully grasped, the corresponding reward is multiplied by 10. All other settings remain consistent with the default configuration in Robosuite.

Similarly, in the TwoArmLift task, the agent must learn to lift a pot-like object with two handles. In this case, the algorithm controls two robotic arms, each with the same configuration as the single-arm tasks. Following the reasons for incorporating dense rewards in the previous tasks, we also introduce dense rewards in this scenario. Specifically, we amplify the grasping reward and reaching reward, setting the grasp reward to 70 and multiplying the proximity reward by $5\times$ to enhance the learning process.

### A.1.4  MOBILE-CATCH TASKS

The task is originated by Zhang et al. (2024). The whole task can be divided into 2 parts, track and catch. The track period is to manipulate the movable robot to reach as close to the object to be thrown later as possible. It usually utilizes the image information while the tactile maps are in absence, so it is not suitable to deploy our algorithm. We use the method proposed in the paper and the pre-trained models to complete the first period.

The latter catching period, in contrast, is contact-rich and relatively fit for our cross modal algorithm. The content of catching period is to literally catch the thrown object and make it stable in the robot's palm. We use environment with same physical parameters. For consistency, we add proprioceptive information presented in original paper along with image and tactile to better evaluate training results. We concatenate image, tactile and proprioception directly.

### A.1.5  BLOCK SPIN TASK

Block spin environment is borrowed from Yuan et al. (2024a), where we add tactile sensors on the robotic hand. The agent is expected to spin the in hand block with an Allegro hand and the overall input is RGB image observation and padded $32 \times 32 \times 3$ tactile sensor data. It is worth noting that the task is different from shadow dexterous hand tasks, in which objects are rotated with a fixed angle in every episode.

### A.2  PSEUDOCODE OF THE PROPOSED SEVERAL CONTRASTIVE METHODS

---
**Algorithm 1** Pseudocode of *soft fusion contrastive learning*

---
**Input**: two modality inputs $q, k$; parameter $K$ for soft fusion contrastive. $k$ is detached for stabilizing the update process.

  1: $logits \leftarrow q \cdot k \,/\, \tau$
  2: $k\_logits \leftarrow k \cdot k$
  3: $k\_index \leftarrow topKmax(k\_logits, K)$
  4: $mask \leftarrow scatter(k\_index)$
  5: $soft \leftarrow softmax(logits) \cdot mask$
  6: $loss \leftarrow -\log(\sum_{i=1}^{N} soft)$

**Output**: soft fusion contrastive loss *loss*

---

### A.3  HYPER PARAMETERS

The important hyper-parameters are shown in Table 3. The detailed information of our CNN encoder is shown below. CNN encoder we adopt corresponds to the designs in drq-v2.

---

**Algorithm 2** Pseudocode of *time contrastive learning*

---

**Input**: two modality inputs $q, k$; parameter $K, t$ for soft fusion contrastive and time contrastive. $k$ is detached for stabilizing the update process.

1: $logits \leftarrow q \cdot k \,/\, \tau$
2: **for** each $i \in [1, N]$ **do**
3: $\quad k\_index_i \leftarrow \bigcup_{j=\min(1, i-t)}^{\max(N, i+t)}\{j\}$ {we use $t$ preceding and succeeding frames}
4: **end for**
5: $mask \leftarrow scatter(k\_index)$
6: $soft \leftarrow softmax(logits) \cdot mask$
7: $loss \leftarrow -\log(\sum_{i=1}^{N} soft)$

**Output**: time contrastive loss *loss*

---

**Algorithm 3** Pseudocode of *ViTaS* guiding reinforcement learning

---

**Initialization**: parameters in policy $\pi_\theta$, CNN encoder and CVAE.
**Notations**: We denote $C$ as numbers of trajectories collected per step, $U$ as update times for policy each iteration in training process.

1: **while** train until reach horizon **do**
2: $\quad$ Run policy $\pi_\theta$ and collect $\{o_i, t_i, a_i, r_i\}$ to buffer $\mathcal{B}$ for $C$ times.
3: $\quad$ Estimate advantages $A_i$ in $\mathcal{B}$ using $r_i, \gamma$.
4: $\quad$ **for** $i \in [1,U]$ **do**
5: $\quad\quad$ Prepare observation $f_o(o_i), f_t(t_i)$ to interact with environment.
6: $\quad\quad$ Set $s_i \leftarrow \{f_o(o_i), f_t(t_i)\}$
7: $\quad\quad$ Calculate policy loss via

$$\mathcal{L}_{PPO} = \min\left(\frac{\pi_\theta(a|s)}{\pi_{\theta_k}(a|s)}A(s,a), \text{clip}\left(\frac{\pi_\theta(a|s)}{\pi_{\theta_k}(a|s)}, 1-\epsilon, 1+\epsilon\right)A(s,a)\right)$$

8: $\quad\quad$ Calculate soft fusion contrastive loss $\mathcal{L}_{CON}$ according to Equation (2) and Equation (3)
9: $\quad\quad$ Calculate CVAE loss $\mathcal{L}_{VAE}$ via Equation (5)
10: $\quad\quad$ Update policy, CNN encoder and CVAE via optimizing $\mathcal{L} = \lambda \cdot \mathcal{L}_{CON} + \mu \cdot \mathcal{L}_{VAE} + \mathcal{L}_{PPO}$
11: $\quad$ **end for**
12: **end while**

---

### A.4 ADDITIONAL BASELINES

Apart from 4 baselines raised above, we add extra visual RL algorithms as baselines. DrQ-V2, a critical visual RL algorithm, is adopted here to further illuminate the merit of ViTaS. Specifically, we use image-only and combined image-tactile observations as inputs respectively to test the efficiency based on 4 aforementioned benchmarks, with results listed below.

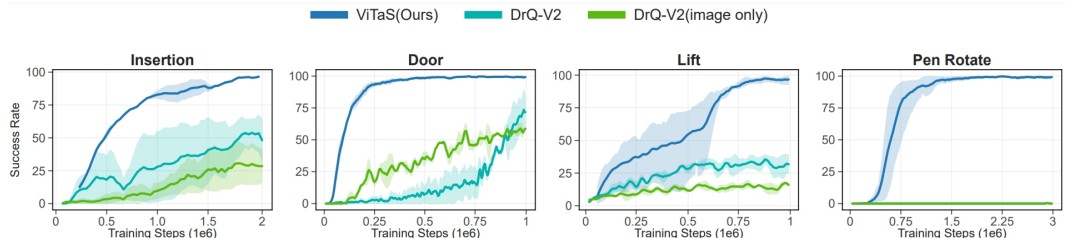

Figure 9: **Learning curve for ViTaS and DrQ-V2**. We validate the performance of image-only and combined image-tactile versions of DrQ-V2 on Insertion, Door, Lift and Pen rotate.

The results in Figure 9 manifest the prominent performance of ViTaS, with a significant improvement compared to DrQ-V2. Despite variance in the policy architecture (off policy like DrQ-V2 vs. on policy like PPO), ViTaS still boasts better feature extraction, leading to a better performance con-

Table 3: Hyper-parameters in ViTaS

| Hyper-parameters | Value |
|---|---|
| RGB Image Input shape | $64 \times 64 \times 3$ |
| Tactile Input shape | $32 \times 32 \times 3$ |
| Soft Contrastive Learning projection dim | 1024 |
| InfoNCE temperature | 0.1 |
| CVAE Condition Projection dim | $81 \times 64 \times 2$ |
| CVAE latent space dim | 256 |
| Policy learning rate | $10^{-4}$ |
| CNNs learning rates | $10^{-4}$ |
| MLP truncate dimension | 64 |
| batch size in training process | 512 |
| $T_{switch}$ | 100 |
| $K$ in soft fusion contrastive | 5 |
| $\gamma$ in PPO | 0.99 |
| clip range in PPO | 0.2 |
| $\lambda$ | 1 |
| $\mu$ | 0.1 |
| batch size in soft fusion contrastive | 4096 |
| $t$ in time contrastive | 5 |
| $horizon$ | $3 \times 10^6$ |
| update per iteration | 10 |

Table 4: CNN parameters in ViTaS

| Layer | in channel | out channel | kernel size | stride |
|---|---|---|---|---|
| Layer 1 | 3 | 32 | 3 | 2 |
| Layer 2 (ReLU) | - | - | - | - |
| Layer 3 | 32 | 32 | 3 | 1 |
| Layer 4 (ReLU) | - | - | - | - |
| Layer 5 | 32 | 32 | 3 | 1 |
| Layer 6 (ReLU) | - | - | - | - |
| Layer 7 | 32 | 32 | 3 | 1 |
| Layer 8 (for image) | 32 | 64 | 3 | 2 |
| Layer 9 (for image) | 64 | 64 | 4 | 1 |
| Layer 10 (for image, ReLU) | - | - | - | - |
| Layer 8 (for tactile) | 32 | 64 | 1 | 1 |
| Layer 9 (for tactile, ReLU) | - | - | - | - |

sequently. Moreover, the distinction between two versions of DrQ-V2 also shows the critical role tactile information plays in representation learning.

## A.5 VALIDATE ROTATE ENVIRONMENTS

To show we adequately optimize the rotate environments for all algorithms, we conduct additional experiments to validate the environments. Specifically, we train M3L and VTT for up to 20 million timesteps in Block Rotate and Egg Rotate environments.

The results in Figure 10 clearly validate Block Rotate and Egg Rotate. The training results of M3L resonate with Sferrazza et al. (2023), as in gymnasium rotate tasks success demonstration usually

has rewards around $8000$ (Towers et al. (2024)). Moreover, the similarity M3L and VTT shows before 3M in Figure 10 and Figure 5 also implies the accuracy of our experiments before.

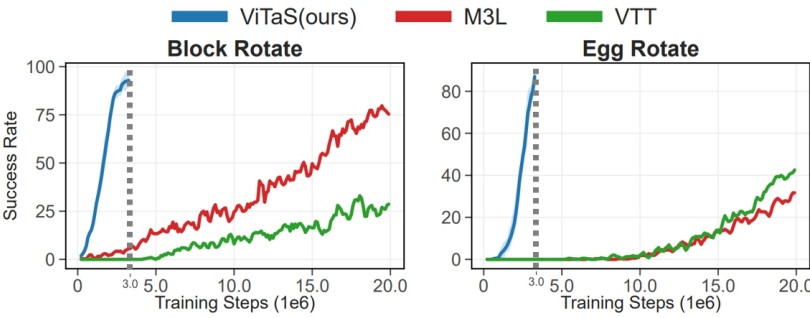

Figure 10: **Extensive learning curve for ViTaS, M3L and VTT**. We validate the rotate environments by enlarging total timesteps of M3L and VTT.

### A.6 Ablation on CVAE

Apart from the experiments presented in Figure 6 (b), we conduct extensive experiments to further investigate how CVAE contribute to our algorithm. Block Rotate and Egg Rotate are added with 3 seeds. Moreover, we shift the shape of objects in Lift task, shrinking the target object into various smaller shapes. The results are shown below.

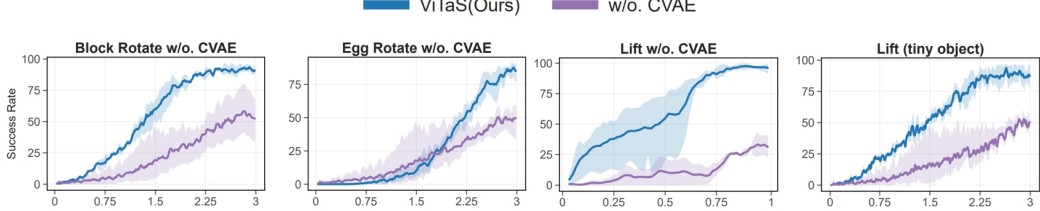

Figure 11: **Extensive experimental results for ViTaS and ViTaS ablating CVAE**. We show the results in Egg Rotate, Block Rotate, Lift and Lift with tiny objects.

From the figure above, it can be deduced that the performance of ViTaS decreases by approximately $35\%$ on average after the removal of the CVAE module. Furthermore, in the task of Lift with tiny object, ViTaS demonstrates remarkable robustness, as its performance at 3M is comparable to that of original Lift task. However, when ablating CVAE module, the performance of ViTaS drops by $50\%$, which underscores the significant role of CVAE in capturing the nuances of visual imagery.

### A.7 Clarification of Contrastive Formula

The original formulation of Equation (2) is like below:

$$\mathcal{L}_{\texttt{CON,1,i}} = -\mathbb{E}\left[\log \frac{\sum_{p \in \mathcal{P}_1(i)} \exp(q_p \cdot q_i \ / \ \tau)}{\sum_{p \in \mathcal{P}_1(i)} \exp(q_p \cdot q_i \ / \ \tau) + \sum_{n \in \mathcal{N}_1(i)} \exp(q_n \cdot q_i \ / \ \tau)}\right] \tag{6}$$

Specifically, we use contrastive learning with multiple positives $q_p$, combining with infoNCE loss. As mentioned above, we use the key featuers to discriminate positives and negatives, so we use encoders to obtain samples in accordance to He et al. (2020) and Chen et al. (2021). Moreover, we

use *cosine similarity* in Sim. The extracted features $f_o(o_i)$ and $f_t(t_i)$, however, are normalized at the very end, leading to $(\sum_{j=1}^{|o_i|} f_o(o_{i,j})^2)^{1/2} = 1$. Thus, the cosine similarity is numerically equal to the corresponding dot product, which is all we need to calculate.

## A.8 RECONSTRUCTION RESULTS FOR CVAE MODULE

To effectively demonstrate the impact of CVAE module, we employ weights from ViTaS in the Egg Rotate task for image reconstruction from pure gaussian noise conditioned on visuo-tactile embedding. We compare the performance under varying levels of Gaussian noise added to the observation space (both visual and tactile) against the token-based MAE method used in M3L.

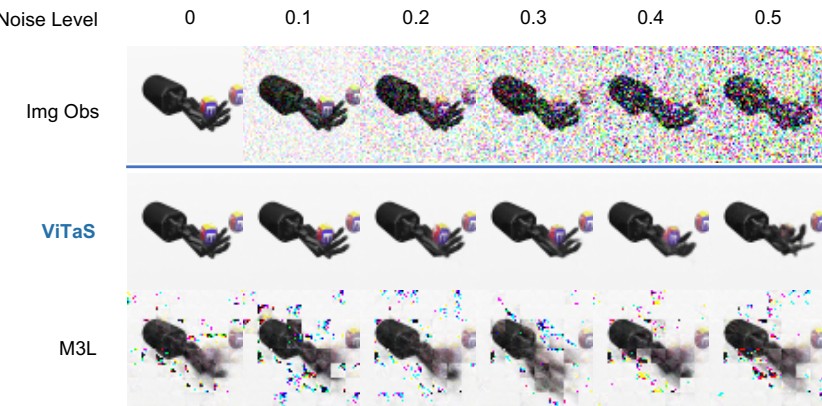

Figure 12: **Reconstruction results under different level of observation noise**.

As illustrated in figure 12, the results indicate that our approach surpasses the token-based MAE in reconstructing critical interaction details, such as finger joint positions and the egg's location, which are vital for the task. Our method also maintain robust under higher level of noise, underscoring the high quality of the visuo-tactile embeddings used as conditions.

Furthermore, we conducted experiments where heavy noise (noise level 0.5) was introduced to either the visual or tactile inputs while keeping the other noise-free. As shown in figure 13,these experiments yielded superior generation performance compared to scenarios with heavy noise in both inputs, demonstrating the complementary nature of the two modalities.

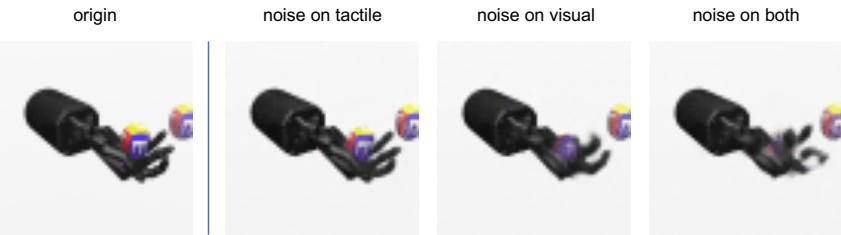

Figure 13: **Reconstruction results under heavy noise applied to different modalities**.

## A.9 t-SNE VISUALIZATION FOR VISUO-TACTILE EMBEDDING

To visually demonstrate the contribution of ViTaS at the feature level, we extracted visuo-tactile embeddings from three models: ViTaS, ViTaS without CVAE, and ViTaS without contrastive learning. These embeddings were obtained along the same egg rotate trajectory and visualized within the same space, as shown in figure 14.

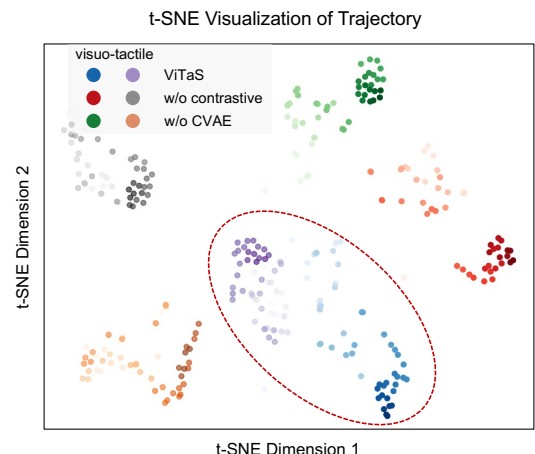

Figure 14: **t-SNE visualization of visuo-tactile feature space**.

In the figure, the visual and tactile representations are color-coded as follows: blue-purple for visuo-tactile embedding of ViTaS, red-gray for ViTaS without contrastive, and green-orange for ViTaS without CVAE. The gradient from light to dark indicates the progression of a trajectory from start to finish. It is evident that in ViTaS, the features from the two modalities exhibit strong complementary characteristics. Additionally, within the same modality, the different phases of the trajectory are spatially separated rather than clustered together. Such feature distribution is advantageous for the downstream policy network, a benefit not observed when using CVAE or contrastive learning alone.

A.10  ABLATION ON HYPER-PARAMETER $\mu$ IN CVAE LOSS

To give a more precise analysis on the value of $\mu$ in CVAE loss presented in Equation (1), we conduct experiments with different value of $\mu$. Specifically, we let $\mu \in \{0.01, 0.1, 1, 10\}$ respectively to show the value we chose in ViTaS (i.e. $\mu = 0.1$) is optimal across different orders of magnitude.

We use Block Rotate and Egg Rotate, two of the toughest tasks among our benchmarks to manifest the superiority over other different values. The settings of all experiments are aligned with all corresponding experiments mentioned in Section 4, and the results are shown below.

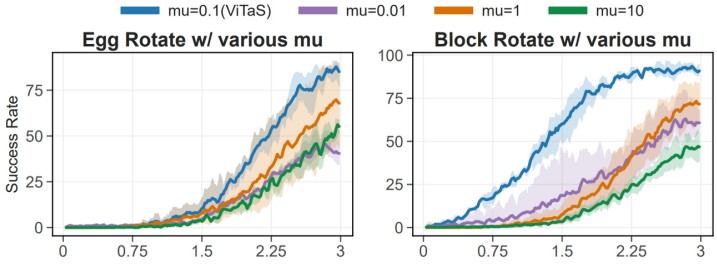

Figure 15: **Performance among different value of $\mu$ in CVAE loss.** We show the success rate in 2 hard tasks with various coefficient in Equation (1).

As discerned from Figure 15, it is evident that when $\mu = 0.1$, the results are far better than other settings of $\mu$, resonating with the hyper-parameter presented in ViTaS. Having obtained the magnitude of $\mu$, we want to further clarify that $\mu = 0.1$ is optimal than other values like $\mu = 0.05$ or $\mu = 0.5$. The overall results are shown in Table 5.

In addition to comparing the performance in tasks with different value of $\mu$, we extract visuo-tactile features from the same successful Egg Rotate trajectory and create the t-SNE plots in order to visu-

Table 5: **Additional ablation on** $\mu$ **in CVAE.** Besides several value of $\mu$ in the figure above, we also evaluate some value near that in ViTaS (i.e $\mu = 0.1$). Green for optimal results while purple for suboptimal.

| $\mu$ | 0.1 (ViTaS) | 0.01 | 1 | 10 | 0.5 | 0.05 |
|---|---|---|---|---|---|---|
| Egg Rotate | **85.1**±**2.3** | 42.5±8.3 | 67.7±10.9 | 58.4±1.2 | 68.6±4.5 | 77.8±3.1 |
| Block Rotate | **93.2**±**2.0** | 60.3±7.4 | 74.1±8.7 | 48.9±6.0 | 80.2±3.6 | 76.5±4.8 |
| Average | **89.3**±**2.0** | 48.7±7.6 | 71.4±8.8 | 53.0±3.3 | 73.9±4.0 | 77.6±3.7 |

ally demonstrate the impact of different $\mu$ values on our algorithm in the feature space. As presented in Figure 16, the patterns for $\mu = 0.01$ and $\mu = 10$ indicate that although the embeddings of visual and tactile observations in latent space could be easily discriminated, the large mean distance between the two modalities suggests poorer extraction capability, given the resemblance between the features of the two modalities at the same timestep in a trajectory. The two patterns for $\mu = 1$ have a better performance than that in $\mu = 0.01$ and $\mu = 10$ since the mean distance between light and dark orange scatter plots is reduced. However, the two patterns seem to be irrelevant to each other and lack of local consistency, which indicates the inability for obtaining the similarity between visual and tactile observations, and the sub-optimality of $\mu$.

For $\mu = 0.1$ (which is used in ViTaS), the adjacent visuo-tactile embeddings form a cohesive whole between two modalities in contrast to other settings above. Within a single modality, the embeddings corresponding to different states are well-separated, allowing the representation to better capture the differences between states. This improved state representation leads to better performance in downstream tasks highlighted in Figure 15 and Table 5, as the policy can distinguish and respond more effectively to different states. Additionally, the cohesive integration of visuo-tactile features allows the model to leverage the fused information from both modalities, further enhancing the overall performance.

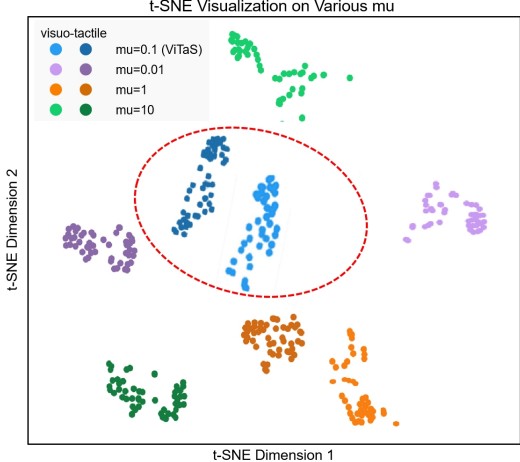

Figure 16: **t-SNE visualization of visuo-tactile feature space for models with various** $\mu$.

Upon examination of Figure 15, Table 5 and Figure 16, we could reach the conclusion that $\mu = 0.1$ is a prominent hyper-parameter for the coefficient of CVAE loss, echoing with the value of hyper-parameter in Equation (1) and Table 3.

