# OpenReview forum: "ViTaS: Visual Tactile Soft Fusion Contrastive Learning for Reinforcement Learning"
_ICLR.cc/2025/Conference — Submitted to ICLR 2025_

### Official Review · Reviewer_TtAv · 2024-11-03

**Soundness:** 3
**Presentation:** 3
**Contribution:** 2
**Rating:** 5
**Confidence:** 4

**Summary:**

This paper proposed a framework ViTaS, a contrastive learning based framework to enhance reinforcement learning for vision-tactile manipulation tasks in simulation. By incorporating a soft vision-tactile contrastive learning term with CVAE supervision, ViTaS helps various PPO-based manipulation tasks achieve better performance.

**Strengths:**

1. Applied cross-modal contrastive learning method to an interesting use case
2. Conducted various experiments with many baselines, showing a significant boost in curves

**Weaknesses:**

1. The manuscript is not well written. There are dozens of cases that
- Lacks coherence. In Sec 2, 3, and 4 it confused me a lot to see methods or results without knowing your motivation for doing so, or how to interpret the results. The entire paragraph from Line 418 doesn't belong here.
- Figures and Tables
  + Many referred in context 2 pages before/after itself while you don't have to.
  + Details in Fig 1 are not explained, making this teaser confusing
  + Fig 2 is not clear enough, especially the soft fusion contrastive learning part. You showed lots of details in different colors, while it looks just like time contrastive to me. Also, what is the graph in the CVAE module visualization?
  + Fig 5, where is the drop you mentioned at Line 366?
  + Fig 6 It's unclear how to evaluate the two settings. i.e. what patterns prove ViTaS is superior?
  + Why do you use the same abbreviation rule for w/ and w/o in Table 2? Also, you mentioned K=1 and K=50, but the K value for other settings is not mentioned until the ablation for K itself.
- Regarding your contributions, from my PoV (1) and (2) are the same thing. Also, the average improvement of 51% is not shown in your experiment section.
- Missing method details. This is probably due to you mimicking Sec 3.2 of "Self-supervised Co-training for Video Representation Learning" since your Sec 3.1 is mainly adopting this method in your scenario, but I strongly suggest you revise your method sections. Your RL setting has more details to tell than copying theirs, as you showed in your pseudocode in the Appendix. For example,
  + Your top-K is found in the buffer, which is never mentioned in context as an important implementation detail.
  + Also, the metric for similarity is non-trivial. You should explain your usage of the encoder in formula (2).
- Typos
  + Line 151, takes information from two modalities, or simply two modalities'.
  + Line 154, cross-modal
  + Line 215, full stop
  + Line 261, involving
  + Line 286, zero-padded
  + Line 431, results are
  + Line 437, is not explained "below"

2. Your entire framework is based on the observation "We note that when two tactile maps are resembling, the key features derived from the corresponding visual data should likewise bear a strong resemblance and vice versa". This is a strong and dangerous claim without clarifying the scope. For more broad vision-tactile manipulation tasks, I can easily construct counter-examples by replacing every part of the grasped object except the contact surfaces, not to mention the backgrounds. In this case, (1) does this claim work only in your task setting, with restricted object types that satisfy this rule? (2) will applying ViTaS in turn harm the generalizability of the RL agent? Please add more discussion on this part, elaborating either on the nature of your dataset or the limitation of your method. Please correct me if I misinterpreted your claim.

**Questions:**

1. In your CVAE ablation you showed only the pen rotation test. From Fig 4 it seems the pen is large enough in the visual image. What about other tasks? For example, can your CVAE work in other experiments where the object of interest is small, or its visual states are subtle? Please show some ablations on this. For the same reason, the explanation in Line 381 "This token-based feature extraction is prone to overlooking critical information compared to pixel-level extraction, thus resulting in the capture of fewer essential features" is not convincing. Can you show some qualitative results?
2. The last two questions in the Weaknesses section

---

> ### Author Response · Authors · 2024-11-21
> **Thank you for your constructive feedback!**
>
> We would like to express our sincere gratitude for the in-depth suggestions and careful readings of the reviewer. Thank you for appreciating our contribution. We have revised the manuscript and will address your questions below.
>
> > Lacks coherence. In Sec 2, 3, and 4 it confused me a lot to see methods or results without knowing your motivation for doing so, or how to interpret the results. The entire paragraph from Line 418 doesn't belong here.
>
> Thanks for your suggestion! We rearrange the writings of Sec 2, 3, and 4 in the new revision for clarity. For example:
>
> 1. We have enhanced the description and explanation in Sec 2 _Contrastive Learning_.
> 2. We substantially revised this section, addressing key points raised by reviewers. Notably, we modify Eq. 2, and provide further explanation on the RL details like topKmax and similarity function we use for better accuracy and clarity.
> 3. We describe the motivation of soft contrastive learning and CVAE at the beginning of Sec 3. Furthermore, we add substantial experiments in Sec 4 and Appendix A.4, A.5, A.6, A.8 and A.9 to show the rationality for our motivation.
> 4. We provide additional details in the experiments. For instance, we add the interpretation of the different channels in the tactile maps. More details are added in Appendix A.1.
> 5. We add more coherence to make the paper more comprehensible.
>
> Moreover, the paragraph in Line 418 is also moved to make the reading easier.
>
> > Many referred in context 2 pages before/after itself while you don't have to.
>
> Thanks for your suggestion! We carefully revise the context according to your suggestion. Specifically,
>
> 1. We have relocated Table 2 to a more prominent position, immediately following the explanation of column names.
> 2. The placement of Fig 7 (which illustrates the visuo-tactile embedding visualization), has been modified to enhance its proximity to the relevant textual references.
> 3. The position of the Fig 5 (Learning curves for all 5 methods in 9 primitive tasks and the derived ones.) has been adjusted to approximate proximity to the referenced location.
> 4. We add more coherence of figure and context. For example, we add the link to Fig. 4 in Sec 4.1, Fig. 5 and teaser in Sec 4.3.
>
> > Details in Fig 1 are not explained, making this teaser confusing.
>
> Thanks for your suggestion! We add more clarification about the average success rate of different algorithms and number of trainable parameters in representation learning in Sec. 1 and Sec. 4 to further explain the teaser.
>
> > Fig 2 is not clear enough, especially the soft fusion contrastive learning part. You showed lots of details in different colors, while it looks just like time contrastive to me. Also, what is the graph in the CVAE module visualization?
>
> Thanks for your suggestion! We have updated Fig 2 to more clearly illustrate the soft contrastive learning component. We would like to reinterpret the soft fusion contrastive part that when one observation is similar to another (indicated by the brown dashed line), the _dual_ samples are considered positive pairs of each other (the dark red dashed line). Conversely, when observations are dissimilar, their corresponding _dual_ samples are negatives. We recall that we denote $\\{o_i, t_i\\}$ at the same timestep as _dual_ samples for each other.
>
> Additionally, soft fusion contrastive could be discerned from time contrastive, since neighboring frames are not always the positives in Fig 2.
>
> We have also revised the graph in CVAE. As elaborated in Sec 3.2, CVAE module uses key features from observations as _condition_ to reconstruct the observation $o_t$ and calculate the MSE as the optimization target.
>
> > Fig 5, where is the drop you mentioned at Line 366?
>
> Thanks for your careful observation! We would like to respectfully reinterpret that when we view the results at **$10^6$** training timesteps, the performance _does_ drop for all algorithms.
>
> We understand it may arise from insufficient emphasis on the numbers along the x-axis, so we also add gray dotted lines in the last 2 plots in Fig 5 for a more lucid explanation.
>
> > Why do you use the same abbreviation rule for w/ and w/o in Table 2? Also, you mentioned K=1 and K=50, but the K value for other settings is not mentioned until the ablation for K itself.
>
> Thanks for your suggestion! In the revised version, we add ‘w/o.’ in the column to distinguish w/ and w/o. . We also add some coherence and clarification for the default value of $K$ in Sec 3.1.
>
> > Regarding your contributions, from my PoV (1) and (2) are the same thing. Also, the average improvement of 51% is not shown in your experiment section.
>
> Thanks for your suggestion! We have added the clarification of the average improvement of $51$% in both Introduction and Experiments section. It is true that contribution (1) and (2) are fairly similar from some perspective, however, we would like to highlight the soft fusion contrastive learning as an important part of ViTaS, so we keep them separate.

---

> ### Author Response · Authors · 2024-11-21
> **Thank you for your constructive feedback!**
>
> > Fig 6 It's unclear how to evaluate the two settings. i.e. what patterns prove ViTaS is superior?
>
> Thanks for your suggestion! We polish this part and add a more specific explanation of the setting and how it proves ViTaS superior according to the t-SNE visualization. Please refer to Fig 7 (previous Fig 6) in Sec 4.4.
>
> > Typos.
>
> Thanks for your careful reading and we are very sorry to make these rookie mistakes. All typos have been corrected immediately and we will be more focused on typos or grammar mistakes.
>
> > Missing method details. This is probably due to you mimicking Sec 3.2 of "Self-supervised Co-training for Video Representation Learning" since your Sec 3.1 is mainly adopting this method in your scenario, but I strongly suggest you revise your method sections. Your RL setting has more details to tell than copying theirs, as you showed in your pseudocode in the Appendix.
> For example, Your top-K is found in the buffer, which is never mentioned in context as an important implementation detail.
> Also, the metric for similarity is non-trivial. You should explain your usage of the encoder in formula (2).
>
> Thanks for your in-depth suggestion! We significantly revise the Method part to show the details of our RL settings according to your kind and in-depth suggestion. Specifically,
> 1. We change the definition formula of $P_1(i)$, introducing the $\text{Sim}(x,y)$ function to refer to the similarity between features $x$ and $y$. We use _cosine similarity_ in implementation. In Appendix A.7, we analyze the relation between cosine similarity and dot product, and we reinterpret the usage of encoders, as the samples we use in contrastive learning are features instead of raw observations.
> 2. We revise the pseudocode of ViTaS (Algorithm 3) significantly, with more accurate formulas and brief statements.
> 3. We add more clarification for the topK used in the pseudocode of ViTaS in Sec 3.2.
>
> > Your entire framework is based on the observation "We note that when two tactile maps are resembling, the key features derived from the corresponding visual data should likewise bear a strong resemblance and vice versa". This is a strong and dangerous claim without clarifying the scope. For more broad vision-tactile manipulation tasks, I can easily construct counter-examples by replacing every part of the grasped object except the contact surfaces, not to mention the backgrounds. In this case, (1) does this claim work only in your task setting, with restricted object types that satisfy this rule? (2) will applying ViTaS in turn harm the generalizability of the RL agent? Please add more discussion on this part, elaborating either on the nature of your dataset or the limitation of your method. Please correct me if I misinterpreted your claim.
>
> Thanks for your careful readings and constructive questions! We would like to respectfully reinterpret that our key insight (a.k.a. "We note that when two tactile maps are resembling, the key features derived from the corresponding visual data should likewise bear a strong resemblance and vice versa") is within _one_ task and _the same_ trajectory. In other words, the key features in visual data of resembling two tactile maps in a certain trajectory bear a strong resemblance in turn. The insight is rather straightforward. As stated in the original paper, the tactile and visual information somehow show similarity, leading to a clear contrastive learning idea.
>
> On the basis of the insight, we can answer the rest of the question. Firstly, the counter-examples are not valid since we only care about the resemblance within _one_ task without changing any settings for objects or backgrounds.
>
> For question (1), we conduct extra experiments again including more object settings like shape or size, and the results are shown in the revised manuscript.
>
> For question (2), as the key insight only within one task, ViTaS _could not_ harm the generalizability of the RL agent theoretically, and both our original experiments in Fig 2. and ablation study echoes with the claim.
>
> However, ViTaS may struggle with representations under significant noise and scalability to more complex manipulation tasks, which are also tough issues for the RL community. We believe that ViTaS may offer some insights to pave the way to solve the problems.

---

> > ### Author Response · Authors · 2024-11-21
> > **Thank you for your constructive feedback!**
> >
> > > In your CVAE ablation you showed only the pen rotation test. From Fig 4 it seems the pen is large enough in the visual image. What about other tasks? For example, can your CVAE work in other experiments where the object of interest is small, or its visual states are subtle? Please show some ablations on this.
> >
> > Thank you for the suggestion! We conduct extensive experiments in the ablation of CVAE. Besides Pen Rotate w/o. CVAE, we also conduct experiments on Block Rotate w/o. CVAE, Egg Rotate w/o. CVAE and Lift w/o. CVAE. Moreover, to show CVAE could help recognize subtle visual states, we also conduct the Lift task with a tiny object. The results have been shown in Fig 11 in Appendix A.6.
> >
> > The results reveal the great contribution CVAE makes in ViTaS, especially to discriminate visual details and extract key features.
> >
> > > For the same reason, the explanation in Line 381 "This token-based feature extraction is prone to overlooking critical information compared to pixel-level extraction, thus resulting in the capture of fewer essential features" is not convincing. Can you show some qualitative results?
> >
> > We appreciate the opportunity to elaborate on the comparative analysis between ViTaS and the M3L baseline, which is based on MAE token reconstruction.
> >
> > In Fig 12, we provide a visual comparison of the reconstruction quality and noise robustness between ViTaS and M3L. Our results demonstrate that ViTaS significantly outperforms M3L, particularly in accurately reconstructing joints and interactions. ViTaS maintains superior reconstruction quality even under higher noise levels. In contrast, M3L's token-level reconstruction appears coarser, capable of capturing the general structure but lacking in the precise details crucial for dexterous manipulation, such as joint positions. This highlights a deficiency in M3L's latent space, where critical manipulate-related information is lost.
> >
> > We believe these findings underscore the advantages of our pixel level reconstruction approach, CVAE in capturing important detailed information for robotics manipulation. Thank you for your valuable insights.

---

> > > ### Author Response · Authors · 2024-11-25
> > > **Thank you for your time and thoughtful review of our work!**
> > >
> > > Dear reviewer TtAv, as the rebuttal discussion is ending soon, we want to check if our response has addressed your concerns. We sincerely hope that our replies have clarified the issues, and we would greatly appreciate it if you could consider raising your score. Thank you again for your time and thoughtful review of our work.

---

> > > > ### Comment · Reviewer_TtAv · 2024-11-27
> > > >
> > > > Thank you for your response. Most of my concerns are properly addressed. Even though, I am not convinced by your assumption "within one task and the same trajectory". This still poses strict restrictions to the potential use cases of ViTaS which also limits the contribution. I will raise my rating for the presentation improvement for now.

---

> ### Author Response · Authors · 2024-12-01
> **Thanks for your positive feedback and constructive suggestions!**
>
> > Even though, I am not convinced by your assumption "within one task and the same trajectory". This still poses strict restrictions to the potential use cases of ViTaS which also limits the contribution.
>
> Thank you for your positive feedback and constructive suggestions! We apologize for any confusion regarding the scope of our method. To show that our method can be used in a scope as wide as other mainly RL methods, we restate our claim “_one task same trajectory_” with a clearer explanation, including the characteristics of human operations combining vision and touch in real life, the theoretically task-specific RL training paradigm, and the generalization ability shown in our experiments:
>
> 1. **A real-life example to re-clarify our method.**  For example, when we try to insert a plug into a socket, we use both of our eyes to see where we are now, and our fingers to feel the force in different plug stages. Since we only do one thing at a time, the tactile we feel and the observation we see is just about this task specific trajectory, and there’s no need for us to remember this feeling when doing a totally different thing
> like spinning a block or turning a doorknob. However, this mode of information processing is useful for handling similar problems. For instance, in the same information processing method, we can insert a three-prong plug after learning to insert a two-prong plug, and we can also insert a circle-shaped plug after learning to insert a rectangle-shaped plug. This phenomenon implies that an agent could accomplish other similar but slightly different tasks after learning one task, also reflected in our approach.
>
> 2. **Technically answer for your concerns.** Concerning your question about the limitation imposed by "within one task and the same trajectory," we would like to re-clarify the points. This expression might have led to a misunderstanding of the method's applicability. However, we explain that it can be applied broadly.
> - **Re-clarify _one task_**. Our method has been trained following the **task-specific** RL paradigm, the same as any other methods in this area without more additional limitations, which means that doing multitask-learning or things like that is inherently a different topic. Additionally, as in our real-life example above, inserting a plug into a socket is a single task, and the process is within the same trajectory. After we learn plug insertion, we do not necessarily know how to rotate a pen. From the RL perspective, we don't train a policy for completely different tasks simultaneously. Moreover, within the context of training for a single task, we can have multiple trajectories and scenarios (for example, different shapes of objects in Lift task).
> - **Re-clarify _same trajectory_.**  In soft fusion contrastive, the same pair of positive samples is obtained in the same trajectory for consistency, while different pairs of positive samples are not necessarily obtained from the same trajectory. That’s because all we need is the similar observations as pairs of positive samples, which is not necessarily obtained from the same trajectory (same pair of positive samples is from same trajectory though). As trajectories may reflect different processes to solve the tasks, the learning results are capable of solving the tasks with generalizability and robustness, which will be clarified below.
> - **Experimental validation.** In our paper, we demonstrate through experiments that our method can incorporate random information during training or testing to achieve generalization. This means ViTaS is **not limited to strictly identical trajectories or tasks, and the object shapes can vary**, which shows the potential of ViTaS in a larger scope. For example,
>
>   (1). "Lift-Capsule" and “Lift-Can” in **Fig 5**, we show that the model trained with cube-shaped objects for lifting can effectively perform with capsule-shaped and can-shaped objects during testing.
>
>   (2). "Insertion" in **Fig 5**, we use different shapes of holes and corresponding pegs during both training and testing, allowing the same model to handle various shapes such as triangles, circles, and rectangles.
>
>   (3). “Pen Rotate w/ randomization” in **Table 1**. We randomize the target positions of the pen in the "Pen Rotate" task during both training and testing, meaning the task goals and trajectories are not strictly the same each time. Our model still demonstrates superior performance under these conditions.

---

> ### Author Response · Authors · 2024-12-01
> **Thanks for your positive feedback and constructive suggestions!**
>
> 3. **Towards larger scope of experiments.** We would like to respectfully point out that ViTaS _could_ accomplish other tasks theoretically if we set the training process for the task. As shown in “Re-clarify _one task_” above, ViTaS is trained following a task-specific paradigm. Therefore, if we train the policy properly, our method is capable of accomplishing new tasks successfully as we have tremendously conducted experiments to prove our claim. Moreover, as shown in “Experimental validation” above, ViTaS also owns generalizability besides success in original tasks. Therefore, it is reasonable for us to believe the potential of ViTaS for additional tasks.
>
> We hope our response clarifies the previous assumptions and explains why this does not limit our contribution to a narrow scope. We are looking forward to an engaging discussion!

---

> > ### Author Response · Authors · 2024-12-02
> > **Thank you for your time and thoughtful review of our work!**
> >
> > Dear reviewer TtAv, as the rebuttal discussion is ending soon, we want to check if our response has addressed your concerns.
> >
> > Based on your constructive feedback, we have updated more accurate and detailed clarification of our assumption with a real-life example, further explaining the large potential of ViTaS.
> >
> > We sincerely hope that our replies have clarified the issues, and we would greatly appreciate it if you could consider raising your score. Thank you again for your time and constructive suggestions of our work!

---

### Official Review · Reviewer_385j · 2024-11-03

**Soundness:** 3
**Presentation:** 3
**Contribution:** 2
**Rating:** 6
**Confidence:** 5

**Summary:**

This paper presents ViTaS, a framework that combines visual and tactile data to improve robotic manipulation. It employs Soft Fusion Contrastive Learning and a CVAE module to effectively fuse and reconstruct these modalities. Experiments are done on a diverse set of none tasks in five different benchmarks.

**Strengths:**

- The paper is well-written, organized, and straightforward.
- The experiments are conducted across a diverse set of simulated environments, providing robust validation.

**Weaknesses:**

- **Lack of Motivation:** The authors incorporate a CVAE as a key component in their method, yet the rationale for this choice remains unclear. Section 3.2 includes a brief explanation, but it lacks depth and fails to adequately justify the use of CVAE. The authors should provide a more explicit motivation, detailing why CVAE combined with Contrastive Learning is preferable to the MAE objective used in [1].

- **Contrastive Objective:**  The theoretical differences between the contrastive loss used in this work and that in [2] requires further clarification. Providing a mathematical comparison between the two approaches could offer valuable insight into the unique contributions of this method, particularly regarding distinctions in formulation and the expected effect on learning outcomes.

- **Reconstruction Objective and Noise:** It is surprising that the authors opted for a pixel-based reconstruction objective. Prior research [2,3,4] has consistently shown that pixel-level reconstruction objectives often underperform in noisy conditions, as they may capture irrelevant elements from the observation. How would this approach fare in real-world scenarios with noise-prone tactile sensory data? I recommend that the authors demonstrate the model's robustness by presenting results on noisy or irrelevant observation components, like Gaussian noise, salt-and-pepper noise (similar to [1]), or missing pixels.

- **Limitations:** The limitations specific to this method are insufficiently addressed. The authors should explicitly discuss these limitations in the paper. For instance, addressing scalability to complex manipulation tasks, representation robustness under noise, and real-world applicability could provide readers with a clearer understanding of the practical constraints and areas for future development.

- **Performance on Rotate Tasks:** ViTaS significantly outperforms all other methods on the rotate tasks. The only provided explanation is that these tasks are “contact-rich and require methods to incorporate visual and tactile information jointly.” However, this characteristic seems relevant to all tasks in general. The authors should offer more insights into why ViTaS performs exceptionally well in the rotate tasks specifically.

- **Clarification Needed:** In Section 4.3, the authors claim, “This token-based feature extraction is prone to overlooking critical information compared to pixel-level extraction, thus resulting in the capture of fewer essential features.” This assertion seems counterintuitive, given that ViTs employ an attention mechanism that typically compensates by aggregating information across tokens, effectively capturing both local and global features. The authors should provide references to support this statement.

- **Experiments:** Although the method has been validated across a range of simulated environments, it has not been empirically tested or combined with tactile sensors in real-world experiments. Real-world tactile data is inherently different and would serve as a valuable addition to the validation. However, this is not a major concern, considering the wide range of simulated experiments.

**References**

[1] Sferrazza, Carmelo, et al. "The power of the senses: Generalizable manipulation from vision and touch through masked multimodal learning." (2023).

[2] Zhang, Amy, et al. "Learning Invariant Representations for Reinforcement Learning without Reconstruction."International Conference on Learning Representations.

[3] Bai, Chenjia, et al. "Dynamic bottleneck for robust self-supervised exploration."Advances in Neural Information Processing Systems 34 (2021): 17007-17020.

[4] JDeng, Fei, Ingook Jang, and Sungjin Ahn. "Dreamerpro: Reconstruction-free model-based reinforcement learning with prototypical representations."International conference on machine learning. PMLR, 2022.

**Questions:**

Why is only the visual observation reconstructed in the CVAE, and not the tactile data?

The other Questions are in Weaknesses.

---

> ### Author Response · Authors · 2024-11-23
> **Thank you for your constructive feedback!**
>
> We would like to express our sincere gratitude for the in-depth suggestions and careful readings of the reviewer. Thank you for appreciating our contribution. We have revised the manuscript and will address your questions below.
> > Lack of Motivation: The authors incorporate a CVAE as a key component in their method, yet the rationale for this choice remains unclear. Section 3.2 includes a brief explanation, but it lacks depth and fails to adequately justify the use of CVAE. The authors should provide a more explicit motivation, detailing why CVAE combined with Contrastive Learning is preferable to the MAE objective used in [1].
>
> Thanks for your suggestion! We explain the rationale for using CVAE module by examining visuo-tactile embedding distribution and image reconstruction performance:
> 1. Visuo-Tactile Embedding Distribution: As shown in Appendix A.9, we have visualized the visuo-tactile representations extracted from the same Egg Rotate trajectory using $3$ models: ViTaS, ViTaS w/o. CVAE, and ViTaS w/o. soft fusion contrastive learning. These embeddings are projected into the same space for comparison. The visualization clearly demonstrates that the combination of CVAE and  soft fusion contrastive allows for a more complementary integration of the two modalities, effectively forming a cohesive input for downstream tasks.
> 2. Reconstruction Performance: In terms of reconstruction, we compare our approach with M3L, as detailed in Appendix A.8. Fig 12 illustrates the reconstruction performance of ViTaS and the MAE method used in M3L under varying levels of Gaussian noise in visual and tactile observations. The results show that CVAE significantly outperforms M3L in noise robustness and in capturing interaction-related details such as joint of the robotic hand and the rotated egg. This indicates that features extracted by CVAE have richer interaction details compared to M3L. Additionally, Fig 13 compares different noise injection strategies, revealing that reconstruction is more effective when significant noise is added to one modality while keeping the other noise-free, rather than applying strong noise to both. This highlights CVAE’s ability to better integrate information from different modalities, even when tactile data is not reconstructed but used as part of the conditioning.
>
> > Reconstruction Objective and Noise: It is surprising that the authors opted for a pixel-based reconstruction objective. Prior research [2,3,4] has consistently shown that pixel-level reconstruction objectives often underperform in noisy conditions, as they may capture irrelevant elements from the observation. How would this approach fare in real-world scenarios with noise-prone tactile sensory data? I recommend that the authors demonstrate the model's robustness by presenting results on noisy or irrelevant observation components, like Gaussian noise, salt-and-pepper noise (similar to [1]), or missing pixels.
>
> Thanks for your suggestion! While it is true that simple pixel-level reconstruction can be sensitive to noise, our approach addresses this in several ways.
>
> Firstly, the embeddings used as reconstruction conditions are derived from soft fusion contrastive learning, which provides superior characteristics compared to using contrastive learning alone, such as being more structured and facilitating complementary integration between the two modalities (see Fig 7 and Fig 14).
>
> Secondly, in response to your feedback, we have incorporated Gaussian noise $ \text{noise} \sim \mathcal{N}(0, 0,3) $ into our experiments. As shown in Fig 5, _insertion with noise_ demonstrates that all methods experience a decrease in success rate at corresponding steps when noise is added. However, ViTaS shows the least decline and converges to over $90$% after $3$ million steps, whereas other methods remain below $65$%, highlighting the robustness of our CVAE combined with contrastive learning structure under noisy conditions.
>
> Lastly, as illustrated in Appendix A.8 Fig 12, models trained in a noise-free environment were compared under varying levels of Gaussian noise. The results show that our pixel-level reconstruction outperforms the token-based MAE method used in M3L, exhibiting robustness to noise and retaining the advantages of pixel-level detail without losing crucial information related to joints and interactions, which are vital for task success.
>
> Based on these $3$ points, we confidently address this concern.

---

> ### Author Response · Authors · 2024-11-23
> **Thank you for your constructive feedback!**
>
> > Contrastive Objective: The theoretical differences between the contrastive loss used in this work and that in [2] requires further clarification. Providing a mathematical comparison between the two approaches could offer valuable insight into the unique contributions of this method, particularly regarding distinctions in formulation and the expected effect on learning outcomes.
>
> Thanks for your suggestion! Though DBC presented in [2] shares certain similarity with ViTaS, the two methods have notable disparities, which we will show in a more intuitive manner. We will show the $3$ main difference between the two.
>
> Initially, the inputs for DBC are sole state observations, while for ViTaS are visual and tactile observations. The multi-modal inputs indicate the domain gap between two methods, **as difference (i)**.
>
> Subsequently, ViTaS is composed of two main components: soft fusion contrastive and CVAE. On the one hand, CVAE reconstructs the visual inputs when the visual-tactile feature extracted by the separate encoders is employed as _condition_.  The utilization of CVAE in ViTaS introduces an element of generative modeling that is not present in DBC, thereby highlighting a method divergence between the two, **as difference (ii)**.
>
> On the other hand, the soft fusion contrastive component of ViTaS shares some similarity with DBC in idea level. Now, we will analyze the significant difference between them.
>
> > **Theorem 1**
> > Let $\mathcal{met}$ be the space of bounded pseudometrics on $S$ and $π$ a policy that is continuously improving. Define $\mathcal{F}$ : $\mathcal{met} \rightarrow \mathcal{met}$ by
> > $\mathcal{F}(d,\pi)\left(s_i,s_j\right)=(1-c)\left|r_{s_i}^\pi-r_{s_j}^\pi\right|+cW(d)\left(P_{s_i}^\pi,P_{s_j}^\pi\right)$
> >
> >Then $\mathcal{F}$ has a least fixed point $\tilde{d}$ which is a $π^{∗}$ -bisimulation metric.
>
> According to Theorem 1 [2], the main motivation of DBC is that when two states $s_i, s_j$ are close, the corresponding _optimal action_ should likewise be resembling. In other words, DBC focuses on the interaction of the environment, training the models with the target of similar action when the states are similar.
>
> ViTaS uses the internal similarity between different modalities of states. The motivation of soft fusion contrastive learning in ViTaS lies that when two samples in one modality are analogous, the key features in the _dual_ samples should likewise bear a strong resemblance, which serves as a learning outcome for soft fusion contrastive learning.
>
> DBC primarily relies on the assumption that similar states correspond to similar actions, allowing the model to implicitly learn the transition process. In contrast, our approach with soft fusion contrastive learning differs in that we do not assume that similar visual and tactile modalities will inherently have similar features. Instead, we leverage the characteristic that visual-tactile pairs correspond to the same state, optimizing within their respective spaces to achieve a more effective representation of the same state. This approach ensures that the extracted features are more representative and robust, enhancing the model's ability to capture the nuances of the state accurately. Given the disparities between soft fusion contrastive and bisimulation in DBC, **the difference (iii) has also been analyzed**.
>
> Generally, we analyze the $3$ difference between two methods theoretically, and show the unique contribution of ViTaS. We hope the analysis above could help solve your problems.
>
> > Limitations: The limitations specific to this method are insufficiently addressed. The authors should explicitly discuss these limitations in the paper. For instance, addressing scalability to complex manipulation tasks, representation robustness under noise, and real-world applicability could provide readers with a clearer understanding of the practical constraints and areas for future development.
>
> Thanks for your suggestion! We have added the _Limitation_ part in the revised version. We clarify the robustness of ViTaS to noise in Fig 12, showing the capability to reconstruct visual observations even in significant noise conditions. We discuss two defects in ViTaS: scalability to more complex manipulation and real-world applicability. However, we point out that real-world transfer for ViTaS seems promising since we conduct experiments on a wide range of tasks in various simulated environments, and ViTaS, as mentioned above, shows prominent robustness to noise which is an important hindrance in the real world.

---

> > ### Author Response · Authors · 2024-11-23
> > **Thank you for your constructive feedback!**
> >
> > > Performance on Rotate Tasks: ViTaS significantly outperforms all other methods on the rotate tasks. The only provided explanation is that these tasks are “contact-rich and require methods to incorporate visual and tactile information jointly.” However, this characteristic seems relevant to all tasks in general. The authors should offer more insights into why ViTaS performs exceptionally well in the rotate tasks specifically.
> >
> > Thanks for your suggestion! We will discuss the reasons from $5$ perspectives.
> > 1. Subtle difference among tasks. Many tasks like Insertion and Lift are short-horizon tasks, where tactile information is used in short terms and makes a difference in a few _key time points_. In the Rotate Tasks, however, tactile information is long-horizon and contact-rich to guide the agent, which owns subtle difference among other tasks.
> > 2. ViTaS outperforms significantly in Block Spin besides Rotate Tasks. Block Spin also belongs to long-horizon contact-rich tasks, for the agent needs to rotate the block in a respectively long horizon (about $500$ timesteps). The performance of ViTaS in Block Spin is also $2$ to $3$ times that of baselines (reward is normalized to illustrate conveniently).
> > 3. M3L _could_ train successfully. As shown in Fig 10, when training 20M timsteps in Block Rotate, the success rate of M3L could reach $\sim 75$%, while sample efficiency and performance are far worse than ViTaS.
> > 4. We think using a unified encoder for image and tactile information presented in M3L is not a good idea. Firstly, the inherent domain gap between image and tactile maps implies a separate processing (for example, the $3$ channels in image stands for RGB, while for shear force and normal force maps in tactile). Secondly, in ablation study, the encoder is changed to a unified encoder. Significant drop of performance is viewed (U-column in Table 2), which also proves our claim to some extent. Last but not least, the tactile map is highly dynamic and sparse (with very high values in small regions and nearly zero elsewhere), lacking temporal and spatial coherence, so the reconstruction is no easy task. Moreover, the tactile information is more prone to suffer from noise, making the reconstruction more difficult. Given the $3$ reasons, we think the poor performance of M3L in Rotate is reasonable.
> > 5. Other baselines like VTT uses wrist-mounted force-torque sensor to obtain tactile information, which is a $1 \times 6$ wrist reaction wrench after projection. The low-dimension, compared to $32\times 32\times 3$ tactile information in ViTaS and M3L, is less accurate for Rotate Tasks, leading to poor performance in these tasks.
> >
> > > Clarification Needed: In Section 4.3, the authors claim, “This token-based feature extraction is prone to overlooking critical information compared to pixel-level extraction, thus resulting in the capture of fewer essential features.” This assertion seems counterintuitive, given that ViTs employ an attention mechanism that typically compensates by aggregating information across tokens, effectively capturing both local and global features. The authors should provide references to support this statement.
> >
> > Thanks for your suggestion! In Appendix A.8, we have included a comparison of the reconstruction performance between pixel-level ViTaS and the MAE-based token-level M3L under different noise conditions. The results demonstrate that ViTaS consistently outperforms M3L in both detail and overall reconstruction quality, regardless of the presence of noise. The M3L reconstructions exhibit noticeable grid artifacts due to its token design and fail to preserve crucial information about the positional relationships between joints, fingertips, and the egg during interactions, maintaining only a rough structure. This issue becomes more pronounced as noise levels increase.
> >
> > Moreover, as implied in [5], token-based reconstruction like MAE is rare in visual RL tasks.
> >
> > > Experiments: Although the method has been validated across a range of simulated environments, it has not been empirically tested or combined with tactile sensors in real-world experiments. Real-world tactile data is inherently different and would serve as a valuable addition to the validation. However, this is not a major concern, considering the wide range of simulated experiments.
> >
> > Thanks for your suggestion! We validate the fidelity of our method in a wide range of simulated experiments (more experiments are also added in the revised version). Given the prominent success rate and sample efficiency, robustness to visual and tactile noise compared to M3L in reconstruction (Fig 12) and generalization ability to random setting parameters, we believe that the success of ViTaS could pave the way for real-world experiments, which is reserved for future work.

---

> > > ### Author Response · Authors · 2024-11-23
> > > **Thank you for your constructive feedback!**
> > >
> > > > Q1.Why is only the visual observation reconstructed in the CVAE, and not the tactile data?
> > >
> > > Thanks for your constructive question! We address this issue from $3$ perspectives:
> > > 1. Analysis of M3L's Performance in Rotate Environment: We have analyzed why M3L struggles in the Rotate task, and the answer may lie in that it attempts to reconstruct tactile information. In this setting, tactile sensor data is highly dynamic and sparse (with very high values in small regions and nearly zero elsewhere), lacking temporal and spatial coherence, which complicates training. In contrast, visual observations are more coherent, and using tactile data as a condition for reconstruction allows for implicit integration of this information.
> > > 2. Practical Noise in Tactile Sensors: Beyond the aforementioned reasons, tactile sensors are more prone to noise in practical applications compared to visual data, exacerbating the challenges of reconstruction as discussed in point two.
> > > 3. Reconstruction Experiments: As detailed in Appendix A.8, Fig 13, we have conducted reconstruction experiments with CVAE under $3$ different noise conditions: high noise in visual + no noise in tactile, high noise in tactile + no noise in visual, and high noise in both. The accuracy of joint and interaction information is better in the first two scenarios compared to the third, indicating that solely reconstructing the visual modality instead of reconstructing visual and tactile images can effectively capture the information from both modalities.
> > >
> > > **References**
> > >
> > > [1] Sferrazza, Carmelo, et al. "The power of the senses: Generalizable manipulation from vision and touch through masked multimodal learning." (2023).
> > >
> > > [2] Zhang, Amy, et al. "Learning Invariant Representations for Reinforcement Learning without Reconstruction."International Conference on Learning Representations.
> > >
> > > [3] Bai, Chenjia, et al. "Dynamic bottleneck for robust self-supervised exploration."Advances in Neural Information Processing Systems 34 (2021): 17007-17020.
> > >
> > > [4] JDeng, Fei, Ingook Jang, and Sungjin Ahn. "Dreamerpro: Reconstruction-free model-based reinforcement learning with prototypical representations."International conference on machine learning. PMLR, 2022.
> > >
> > > [5] Li, Xiang  et al. “Does self-supervised learning really improve reinforcement learning from pixels?” (2022).

---

> ### Comment · Reviewer_385j · 2024-11-23
> **Comments on the Rebuttal**
>
> I appreciate the effort from the authors in addressing my concerns. Thank you for your thorough responses, additional experiments and related modifications to the paper.
>
> Among all, your reconstruction results in Fig. 12 are quite impressive and addresses my concern. I believe all the concerns have been adequately addressed, except for one-specifically, the use of CVAE. The authors empirically demonstrate its role via the t-SNE plot in Appendix A.9; however, I still find the intuitive reasoning behind its inclusion unclear. Furthermore, when the authors state, "This highlights CVAE’s ability to better integrate information from different modalities", I can observe this in Fig. 14. Yet, doesn’t the heavy lifting for fusion primarily come from Soft Fusion CL?
>
> I also have one question: \
> In Fig. 12, did the authors replace only the CVAE module with MAE, or did they replace the CVAE+Soft Fusion CL modules with MAE (i.e., comparing ViTaS with M3L)? My confusion arises from Appendix A.8, which states "Reconstruction Results from CVAE", whereas Fig. 12 compares ViTaS and M3L, which is not equivalent since ViTaS also incorporates Soft Fusion CL.
>
> I am willing to increase my score if my questions are adequately addressed.

---

> ### Author Response · Authors · 2024-11-24
> **Thank you for your timely and constructive feedback!**
>
> We would like to express our sincere gratitude for the timely reply, careful readings and positive feedback of the reviewer. Thank you for appreciating our contribution! Now, we will address your questions below.
>
> > Specifically, the use of CVAE. The authors empirically demonstrate its role via the t-SNE plot in Appendix A.9; however, I still find the intuitive reasoning behind its inclusion unclear.
>
> Thanks for your constructive question! The intuitive motivation of CVAE lies in the insight that visual and tactile information often exhibit significant complementarity from humans’ perspective [6], so the integration of visual and tactile information should provide a complete representation for the whole observation, where CVAE walks in with reconstruction.
> The intuitive motivation also echoes with Fig 14, as visual and tactile embeddings show complementarity in the t-SNE figure.
>
> > Furthermore, when the authors state, "This highlights CVAE’s ability to better integrate information from different modalities", I can observe this in Fig. 14. Yet, doesn’t the heavy lifting for fusion primarily come from Soft Fusion CL?
>
> Thanks for your question! We believe that CVAE integrates well _with_ soft fusion contrastive learning to enhance representation learning. As [7,8,9,10,11] stated, reconstruction helps learn representation to guide reinforcement learning. This is evident in Fig 14, where ViTaS demonstrates superior integration compared to models with ablated contrastive and CVAE modules. The w/o. contrastive and w/o. CVAE models show more dispersed feature distributions, indicating poor integration of the two modalities (and _neither_ exhibits clear superiority in this aspect). Therefore, the enhanced multimodal representation in ViTaS can be attributed to the _combined effect_ of both modules.
>
> > Q. In Fig. 12, did the authors replace only the CVAE module with MAE, or did they replace the CVAE+Soft Fusion CL modules with MAE (i.e., comparing ViTaS with M3L)? My confusion arises from Appendix A.8, which states "Reconstruction Results from CVAE", whereas Fig. 12 compares ViTaS and M3L, which is not equivalent since ViTaS also incorporates Soft Fusion CL.
>
> Thanks for your constructive question and careful readings! Your concern is reasonable, and we would like to further clarify our experimental setup. The CVAE reconstruction involves using weights from the ViTaS model, isolating the CVAE component to perform reconstruction following the VAE inference paradigm (sampling from pure Gaussian noise and adding condition to obtain the reconstructed image), where the condition representations are extracted from encoders in ViTaS. The MAE approach involves making patches of visual-tactile observations for reconstruction, with weights derived from the M3L training process.
>
> Since CVAE does not explicitly input the original observations during reconstruction, whereas MAE uses the original visual-tactile observations, having high-quality conditions is crucial for CVAE. Thus, we use ViTaS-extracted visuo-tactile embeddings as condition inputs.  The reconstruction process is tougher for ViTaS, since the reconstructed source in the inference stage is pure Gaussian noise, and the information content (only visual-tactile observation embeddings derived from encoders) does not exceed that used in MAE.
>
> What’s more, the reconstruction capability of the CVAE module itself is not influenced by Soft Fusion CL, as it is evaluated independently. Therefore, although there are inherent differences in the reconstruction approaches of the CVAE and MAE modules, which means it is challenging to compare them by simply swapping the two modules while keeping all other parts unchanged, our visualization method ensures that the comparison is as fair as possible. We hope these explanations address your concerns regarding Fig 12.
>
> **Reference**
>
> [6] Apkarian-Stielau, Patricia, and Jack M. Loomis. "A comparison of tactile and blurred visual form perception." Perception & Psychophysics 18 (1975).
>
> [7] Seo, Younggyo, et al. "Masked world models for visual control." Conference on Robot Learning. PMLR, 2023.
>
> [8] Sferrazza, Carmelo, et al. "The power of the senses: Generalizable manipulation from vision and touch through masked multimodal learning." (2023).
>
> [9] Zhao, Tony Z., et al. "Learning fine-grained bimanual manipulation with low-cost hardware." (2023).
>
> [10] Xu, Zhengtong, et al. "UniT: Unified tactile representation for robot learning." (2024).
>
> [11] Li, Yunzhu, et al. "Connecting touch and vision via cross-modal prediction." Proceedings of the IEEE/CVF Conference on Computer Vision and Pattern Recognition. 2019.

---

> ### Comment · Reviewer_385j · 2024-11-24
> **Comments on the Rebuttal**
>
> I appreciate the authors' detailed responses and their efforts in addressing my doubts. My query regarding the CVAE has been satisfactorily clarified to a significant extent, and my concerns regarding Figure 12 have been resolved.
>
> If future iterations are required, I would suggest that the authors include at least one real-world experiment, as this would shut down the gates for potential concerns and strengthen the work's practical relevance.
>
> Based on the clarifications provided, I have decided to update my score to "Above Threshold".

---

> > ### Author Response · Authors · 2024-11-25
> > **Thanks for your positive feedback!**
> >
> > Dear reviewer, thanks so much for the response and recognizing our efforts! In the future, we will explore the potential of ViTaS to further prove the effectiveness empirically. Thank you again for the timely reply, careful readings and appreciating our works!

---

### Official Review · Reviewer_v2wE · 2024-11-03

**Soundness:** 2
**Presentation:** 2
**Contribution:** 2
**Rating:** 3
**Confidence:** 5

**Summary:**

The authors introduce ViTaS, a framework for integrating visual and tactile information to improve robotic manipulation. ViTaS employs Soft Fusion Contrastive Learning to better fuse these modalities and a CVAE module to leverage complementary visuo-tactile information. The authors conduct some experiments with the proposed method.

**Strengths:**

1. The paper is well-structured and easy to follow.
2. The authors conduct extensive experiments in simulated environments.

**Weaknesses:**

1. Key visual RL baselines, such as DrQ-v2, are missing from the comparisons. The authors should compare their method with DrQ-v2 using both image-only and combined image-tactile observations.
2. More analysis is needed to explain how contrastive learning and the CVAE module contribute to better feature representations.
3. In Figure 5, only the proposed method performs well in the egg, block, and pen rotation environments, while it performs similarly to some baselines in other environments. Are these environments not optimized adequately for the baselines?
4. In Figure 7, the performance variance without CVAE is notably high. Does this indicate that one/multiple of the trials converges to a local optimum? If so, the authors are encouraged to run over more seeds to verify if ViTaS exhibits similar behavior.

**Questions:**

The authors need to address my concerns in the weakness section.

---

> ### Author Response · Authors · 2024-11-21
> **Thank you for your constructive feedback!**
>
> We would like to express our sincere gratitude for the in-depth suggestions and careful readings of the reviewer. Thank you for appreciating our contribution. We have revised the manuscript and will address your questions below.
>
> > Key visual RL baselines, such as DrQ-v2, are missing from the comparisons. The authors should compare their method with DrQ-v2 using both image-only and combined image-tactile observations.
>
> Thanks for your suggestion! We have implemented and incorporated the DrQ-V2 algorithm with image-only and combined image-tactile inputs as new baselines. The performance of the corresponding models was obtained using three different seeds across four tasks. The results have been updated in Appendix A.4 of the revised version, with the updated sections highlighted in blue text.
>
> We observe that although training can be conducted across multiple tasks, the performance of DrQ-V2 is relatively poor. Additionally, by comparing the image-only and combined image-tactile baselines, we have demonstrated the significant role of tactile information on the benchmark.
>
> Further experimental results indicate that simply fusing visual and tactile information does not enhance the performance of RL models in contact-rich tasks. Carefully designed representation learning approaches, such as ViTaS, have shown significant improvement over the baselines in the experiments.
>
> Empirical results of training curves are shown in Fig 9 in Appendix A.4 of revised version.
>
> > More analysis is needed to explain how contrastive learning and the CVAE module contribute to better feature representations.
>
> Thank you for your insightful feedback. We appreciate the opportunity to clarify the contributions of the CVAE and contrastive modules in our model through both feature space analysis and reconstruction visualization.
> 1. Feature Space Analysis:
> - Contrastive Module: In our paper, Fig 7 illustrates the impact of the contrastive module using t-SNE visualization of two similar trajectories with different plug-in shapes. The inclusion of the contrastive module helps structure the features more effectively. The visual and tactile features of these similar trajectories are highly aligned in terms of relative structure and spatial relationships, as evidenced by the nearly parallel visual-tactile lines in the lower left of the figure, with closely corresponding features. In contrast, without the contrastive module, the lines intersect on the right, lacking the consistent structure seen with the contrastive module.
> - CVAE Module: In the appendix, we add Fig 14 and provide a visualization of embeddings extracted from the same trajectory across three models: without CVAE, without contrastive, and our complete model. The CVAE structure aids in distinguishing features at different stages within a single modality, enhancing feature separation.
>
> Overall, as shown in Fig 14, utilizing both modules facilitates a more complementary visuo-tactile information relationship, improving the integration of these features into downstream tasks.
>
> 2. Reconstruction Visualization:
>
> - As detailed in Fig 12 and 13 in appendix, the CVAE structure demonstrates superior reconstruction capabilities from noisy observations compared to M3L, highlighting the high quality of visuo-tactile embeddings used as conditions. Furthermore, reconstruction from single-modality noise is more effective than when both modalities are noisy, indicating a complementary relationship between the modalities. This inter-modality complementarity enhances the model's robustness to noise interference.
>
> We hope these clarifications address your concerns and highlight the effectiveness of our approach. Thank you for your valuable feedback.
>
> > In Figure 5, only the proposed method performs well in the egg, block, and pen rotation environments, while it performs similarly to some baselines in other environments. Are these environments not optimized adequately for the baselines?
>
> Thanks for your suggestion! We would like to respectfully emphasize that some certain baselines like M3L could optimize adequately to the 3 aforementioned Rotate tasks. As the original paper([revised version](https://openreview.net/pdf?id=FMsmo01TaI), Fig 15, P22) pointed out, however, the converged timestep is usually within the range of 15M to 20M, inferior in comparison to ViTaS.
>
> We reproduce M3L with larger timesteps and find the performance is in alignment with the claim of the original paper, further validating the significant improvement in sample efficiency. Moreover, the performance within 3M steps are in great concordance with metrics presented in Fig 2, serving as an additional proof of our experimental procedures in another perspective.
>
> In our revised version, we show the detailed training curve of M3L in Appendix A.5.

---

> > ### Author Response · Authors · 2024-11-21
> > **Thank you for your constructive feedback!**
> >
> > > In Figure 7, the performance variance without CVAE is notably high. Does this indicate that one/multiple of the trials converges to a local optimum? If so, the authors are encouraged to run over more seeds to verify if ViTaS exhibits similar behavior.
> >
> > Thanks for your suggestion! We develop extra $7$ different seeds (a total of $10$ seeds) for ViTaS and ViTaS w/o. CVAE. The outcomes of these experiments have been presented in the subsequent table in the revised manuscript.
> >
> > | Methods      | ViTaS | ViTaS w/o. CVAE     |
> > | :---        |    :----:   |          :---: |
> > | Success rate (%)      | $98 ± 2$       | $71 ± 20$   |
> >
> > The table above shows the mean success rate over a training period of $3\times 10^6$ timesteps. In contrast to the initial ablation study which employs only $3$ seeds, the variance of ViTaS w/o. CVAE drops while still at a high level. We analyze that CVAE endows model the capability to capture detailed information in observations, and empirical results show that the absence of CVAE in ViTaS leads to convergence upon local optimum.
> >
> > We also conduct extra ablation studies with $4$ different settings (Egg Rotate, Block Rotate, Lift, Lift with tiny object) to further elucidate the necessity of CVAE. The results are shown in Fig 11.
> >
> > Generally, these results underscore the pivotal role of CVAE, which also shows in the representation visualization of ViTaS w/o. CVAE in Fig 11, Fig 14 and the second comment above.

---

> > > ### Author Response · Authors · 2024-11-25
> > > **Thank you for your time and thoughtful review of our work!**
> > >
> > > Dear reviewer v2wE, as the rebuttal discussion is ending soon, we want to check if our response has addressed your concerns. We sincerely hope that our replies have clarified the issues, and we would greatly appreciate it if you could consider raising your score. Thank you again for your time and thoughtful review of our work.

---

> ### Author Response · Authors · 2024-12-01
> **Thank you for your constructive feedback!**
>
> We would like to express our sincere gratitude for the in-depth suggestions and careful readings of the reviewer. We have revised the manuscript and will address your questions below.
>
> > I believe the authors need to conduct more experiments and carefully tune the baselines to ensure the results are robust.
>
> Thanks for your suggestion! We would like to respectfully reinterpret the robustness and rationale of our results with more experiments from different baselines.
>
> 1. **Rationale behind the results of M3L.** We understand it may look less convincing that the results of M3L have large variance in Insertion, Lift and Pen Rotate. However, we will show the rationale behind it from $5$ perspectives.
>   - **Alignment in environments.** Nearly half of the tasks in our paper, like Insertion, Door, Pen Rotate, Block Rotate and Egg Rotate, are adopted from M3L without modification. Therefore, the reproduction process of M3L is identical to the original paper, which should be well tuned and ready to reproduce without stressed tuning. Moreover, the average results presented in our paper share a prominent similarity with M3L (for instance, the success rate in Insertion is $74$%, while $\sim 77$% in M3L at $2\times 10^6$ timesteps.), further proving M3L has been well tuned.
>
> - **Comparison of representation capability.** Extra experiments are conducted to show the reconstruction results of ViTaS and M3L with extracted features. As shown in **Fig 12**, the poor reconstruction result shows that M3L is not very excellent at extracting the features given the blurred tokens M3L decoder produces. The inherent capacity for representation extraction within M3L is insufficient. For tasks that demand a higher quality of representation like Egg Rotate in **Fig 12**, the inferior representation significantly impacts the ability of exploration. Consequently, the performance of M3L is of high randomization and sensitive to random seeds, resulting in a less robust performance and high variance.
> - **The inherent design in M3L may cause the large variance.** We argue that the usage of a unified encoder for image and tactile information presented in M3L is not a good idea, and that could be the reason why M3L exhibits large variance. We prove it by adopting a unified encoder for ViTaS in ablation study, which concatenates image and tactile observations and passing through _one shared_ encoder, which has similar designs to M3L. As shown in the first plot in **Fig 6(a)**, ViTaS w/ unified encoder _also_ shows large variance, which surprisingly aligns with results in M3L.
> - **More experiments are conducted.** We have conducted more experiments to increase persuasiveness. For benchmarks in which results have large variance, we use a total of $10$ different seeds to validate the results of **Fig 5**. Specifically, we choose Insertion, Lift, Pen Rotate tasks. The following results are shown at $2\times 10^6, 1\times 10^6, 3\times 10^6$ timesteps respectively, same as those in our paper.
> | Settings  | ViTaS w/ $5$ seeds | M3L w/ $5$ seeds | M3L w/ $10$ seeds    |
> | :----: |  :---: |  :---: | :---: |
> | Insertion (%)   | $99\pm 1$  | $74\pm 20$ | $75\pm 16$  |
> | Lift (%)   | $98\pm 2$  | $50\pm 25$ | $48\pm 16$  |
> | Pen Rotate (%)   | $100\pm 0$  | $30\pm 30$ | $37\pm 22$  |
>
>   As shown in the table, the variance when adopting $10$ seeds drops yet still remains much higher than ViTaS. We argue that the large variance reflects inferiority of M3L in representation learning, echoing with the perspectives above.
> - **Performance in extended horizons.** We have also extended the training horizons for M3L to 20M in Egg Rotate and Block Rotate, two of the hardest tasks as the performance before 3M timesteps is poor for baselines, and the results are shown in **Fig 10**. It is evident that M3L _could_ learn the optimized policy in Egg Rotate and Block Rotate in the process of training though in a poor sample efficiency. It is noteworthy that the results presented in **Fig 10** resonates with those in the original M3L paper([revised version](https://openreview.net/pdf?id=FMsmo01TaI), **Fig 15**, P22), further proving we have tuned the baselines well.

---

> ### Author Response · Authors · 2024-12-01
> **Thank you for your constructive feedback!**
>
> 2. **Rationale behind the results of VTT.** Though directly adopting from the original codebase of VTT, we have still tuned the VTT in several key parameters. We have conducted stressed extra experiments on tuning the VTT.
>
>   - **Learning rate.** We have tuned the learning rate to $5\times 10^{-5}$ and $2\times 10^{-4}$, while the original rate is $10^{-4}$.
>   - **Size of transformer network.** We have tuned the latent transformer architecture, which is the main structure in VTT to fuse the visual and tactile modalities. Specifically, we change the depth of transformer layers from $6$ to $4$ and $8$.
>   - **Coefficient of loss.** The overall loss of VTT could be divided into $2$ main parts, policy loss and representation loss (VTT loss), listed as follows: $$\ell_{model}=c\times\ell_{VTT}+\underbrace{\ell(O_t|z_t^d,a_{t-1})+\ell(r_t|z_t^d,a_{t-1})+\ell_{KL}(q||p)}_{\text{SLAC policy loss}}$$
> The policy is adopted from Stochastic Latent Actor Critic (SLAC) [1], which could be viewed as a whole for the training of policy.
>   We would like to tune the coefficient of VTT loss so that the two components could be more balanced if possible. In the original paper, $c=1$. We have changed into $c=0.1$ and $c=10$.
>
> We have done the tuning experiments on $3$ benchmarks: Insertion, Lift and Block Rotate, with $3$ repeated experiments are conducted with different random seeds. The results are shown below.
>
> Insertion：
> | Timestep      | $0.5\times 10^6$ | $1\times 10^6$| $1.5\times 10^6$    | $2\times 10^6$    |
> | :---        |    :----:   |          :---: | :---: | :---: |
> | VTT   | $30\pm5$  | $52\pm 1$ |$64\pm 1$ |$65\pm 1$ |
> | $lr=5\times 10^{-5}$  | $25\pm3$  | $51\pm 1$ |$53\pm 3$ |$54\pm 2$ |
> | $lr=2\times 10^{-4}$  | $35\pm7$  | $48\pm 5$ |$43\pm 6$ |$50\pm 3$ |
> | $4$ layers of attn  | $25\pm4$  | $50\pm 3$ |$47\pm 2$ |$58\pm 6$ |
> | $8$ layers of attn | $32\pm4$  | $49\pm 4$ |$60\pm 2$ |$59\pm 3$ |
> | $c=0.1$  | $24\pm3$  | $25\pm 10$ |$37\pm 1$ |$38\pm 1$ |
> | $c=10$  | $22\pm3$  | $27\pm 5$ |$36\pm 2$ |$40\pm 1$ |
>
> Lift：
> | Timestep      | $0.25\times 10^6$ | $0.5\times 10^6$| $0.75\times 10^6$    | $1\times 10^6$    |
> | :---        |    :----:   |          :---: | :---: | :---: |
> | VTT   | $31\pm4$  | $50\pm 1$ |$68\pm 10$ |$72\pm 5$ |
> | $lr=5\times 10^{-5}$  | $25\pm4$  | $49\pm 4$ |$58\pm 3$ |$60\pm 1$ |
> | $lr=2\times 10^{-4}$  | $33\pm3$  | $52\pm 10$ |$57\pm 5$ |$59\pm 4$ |
> | $4$ layers of attn | $13\pm4$  | $19\pm 2$ |$26\pm 5$ |$33\pm 3$ |
> | $8$ layers of attn | $27\pm6$  | $47\pm 7$ |$65\pm 4$ |$69\pm 1$ |
> | $c=0.1$ | $27\pm6$  | $35\pm 3$ |$46\pm 4$ |$60\pm 1$ |
> | $c=10$  | $22\pm1$  | $59\pm 5$ |$69\pm 7$ |$67\pm 2$ |
>
> Block Rotate：
> | Timestep      | $0.5\times 10^6$ | $1\times 10^6$| $1.5\times 10^6$    | $2\times 10^6$    | $2.5\times 10^6$    | $3\times 10^6$    |
> | :---        |    :----:   |          :---: | :---: | :---: | :---: | :---: |
> | VTT   | $0\pm0$  | $2\pm 2$ |$0\pm 0$ |$3\pm 1$ |$0\pm 0$ |$1\pm 1$ |
> | $lr=5\times 10^{-5}$  | $0\pm0$  | $0\pm 0$ |$0\pm 0$ |$0\pm 0$ |$1\pm 0$ |$0\pm 0$ |
> | $lr=2\times 10^{-4}$  | $0\pm0$  | $0\pm 0$ |$0\pm 0$ |$0\pm 0$ |$0\pm 0$ |$0\pm 0$ |
> | $4$ layers of attn | $0\pm0$  | $0\pm 0$ |$0\pm 0$ |$0\pm 0$ |$0\pm 0$ |$0\pm 0$ |
> | $8$ layers of attn | $0\pm0$  | $0\pm 0$ |$0\pm 0$ |$0\pm 0$ |$0\pm 0$ |$0\pm 0$ |
> | $c=0.1$  | $0\pm0$  | $0\pm 0$ |$1\pm 1$ |$2\pm 2$ |$0\pm 0$ |$0\pm 0$ |
> | $c=10$  | $0\pm0$  | $0\pm 0$ |$0\pm 0$ |$0\pm 0$ |$1\pm 0$ |$0\pm 0$ |
>
> We could deduce from the table that our current parameters in VTT are relatively well-tuned, showing the rationale of VTT.
>
> 3. **Rationale behind the results of PoE & Concat.** Other baselines like PoE and Concat exhibit much smaller variance in the learning curve and the robustness under different random seeds. Though seeming to be inferior in original VTT paper [2], they astonishingly show excellent performance in Dual Arm Lift, Lift, Lift-Capsule and Lift-Can shown in **Fig 5**,  standing for a brilliant proof of the well-tuned methods which is relatively old but still blooming.
>
> **Reference**
>
> [1] A. X. Lee et al. “Stochastic latent actor-critic: Deep reinforcement learning with a latent variable model” (2020).
>
> [2] Yizhou Chen et al. “Visuo-Tactile Transformers for Manipulation” (2022).

---

> ### Author Response · Authors · 2024-12-01
> **Thank you for your constructive feedback!**
>
> > From my perspective, additional experiments are necessary to clarify why the CVAE approach improves the fusion process. For example, an ablation study on the coefficient of the CVAE loss over the latent space is essential and currently missing.
>
> Regarding your second concern, we have supplemented our paper with a comparison of model performance under different CVAE loss coefficients in **Appendix A.10**, along with a more detailed analysis of how changes in the coefficient affect visuo-tactile representations and the role of this module in feature fusion. The experiment shows that incorporating a well-chosen coefficient for the CVAE can facilitate multimodal representation learning by capturing relevant and complementary features with well-separated states, thereby forming a high-quality fusion of representation.
>
> 1. **Fig 15** and **Table 5** illustrate the model's performance across various CVAE loss coefficient values, $\mu \in \\{0.01, 0.1, 1, 10 \\}$. The value $\mu=0.1$ used in our experiments demonstrates the best performance in terms of learning curves and success rates. Additionally, tests with finer values $\\{0.5, 0.05\\}$ also confirm the appropriateness of $\mu=0.1$, which is why we ultimately select this value as a critical hyper-parameter.
>
> 2. To analyze the impact of CVAE on feature fusion in latent space, we perform t-SNE visualizations of visuo-tactile features obtained under different coefficients, as shown in **Fig 16**. We observe that both excessively small and large coefficients ($\mu=0.01$, $\mu=10$) adversely affected the fusion of visual and tactile modalities. The feature spaces for the two modalities are closer at $\mu=1$ and $\mu=0.1$, while the latter one exhibits a more effective integration of the modalities. To be more specific, the adjacent visuo-tactile embeddings form a cohesive whole between two modalities in contrast to others, and within a single modality, the embeddings corresponding to different states are well-separated, allowing the representation to better capture the differences between states. These advantages at the feature level align with the superior performance observed in **Fig 15** and **Table 5**, underscoring the influence of CVAE in ViTaS.
>
> In summary, we conduct experiments on the CVAE loss coefficient and verify that the choice made in the paper is well-tuned. Through visualization of the visuo-tactile features in the latent space, we demonstrate that larger or smaller CVAE coefficients are detrimental to feature fusion. At the current hyper-parameters we select, the visuo-tactile representations exhibit excellent fused characteristics between two modalities.
>
> We hope our response further clarifies the necessity and unique advantage of CVAE adopted in ViTaS. Thanks again for your constructive questions We are looking forward to an engaging discussion!

---

> > ### Author Response · Authors · 2024-12-02
> > **Thank you for your time and thoughtful review of our work!**
> >
> > Dear reviewer v2wE, as the rebuttal discussion is ending soon, we want to check if our response has addressed your concerns.
> >
> > Based on your constructive feedback, we have updated abundant experiments (like ablation in coefficient of CVAE and tuning of VTT), which we believe could help solve your cocerns.
> >
> > We sincerely hope that our replies have clarified the issues, and we would greatly appreciate it if you could consider raising your score. Thank you again for your time and constructive suggestions of our work!

---

### Official Review · Reviewer_PiGR · 2024-11-04

**Soundness:** 3
**Presentation:** 3
**Contribution:** 3
**Rating:** 8
**Confidence:** 4

**Summary:**

The author introduces a framework to enhance robot manipulation by fusing visual and tactile data. They use a soft fusion contrastive learning approach and want to align these modalities in the latent space to improve RL policy.

**Strengths:**

1. The visualization of t-SNE is good and the overall presentation is easy to read.
2. Extensive experiments and analysis of ablation study, the overall results are convincing.
3. Diverse simulation and environment.

Overall its a good paper to accept.

**Weaknesses:**

1. No real-world experiments were included to validate the effectiveness of the method.

2. Some minor presentation issue should be fixed.

**Questions:**

1. [line281-288]. Please specify what is 32x32x3 tactile sensor and what is 3x3x3. Please specify each channel of them.

2. It would be better for the author to mention what kind of tactile sensors they are using and what kind of sensor signals can be received (eg. normal force, shear force)

3. [line 241,242], there are a lot of papers already showing the dexterous hand's abilities to do in-hand rotation using both vision and tactile or using solely tactile. Thus, the question seems too trivial to answer and cannot be a good point to compare with existing work.

4. To access the generalization of the approach, it would also be good for the author to list the objects used during the testing. Also, It would be better to describe the level of randomization and data distribution for that.

---

> ### Author Response · Authors · 2024-11-28
> **Thanks for your constructive suggestions and appreciation!**
>
> We would like to express our sincere gratitude for the in-depth suggestions and careful readings of the reviewer. Thank you for warmly appreciating our contribution. We have revised the manuscript and followed your suggestions, as stated below.
>
> > 1.[line281-288]. Please specify what is 32x32x3 tactile sensor and what is 3x3x3. Please specify each channel of them.
>
> Thanks for your suggestion! We have updated the details of tactile sensors in the revised version. Specifically, in the $32\times 32\times 3$ tactile sensors, we add the meaning of different channels in tactile maps (shear force in the first and second channels and normal force in the third channel), while in the $3\times 3\times 3$ sensors, the meaning of channels are in alignment with the aforementioned sensors.
>
> > 2.It would be better for the author to mention what kind of tactile sensors they are using and what kind of sensor signals can be received (eg. normal force, shear force)
>
> Thanks for your suggestion! The type of sensor is TactileSim simulated environments presented in [1] and Gelsim 3.0 real-world devices presented in [2], obtaining the tactile maps. As stated in the comment above, tactile map is composed of $3$ channels, the first two of which are shear force while the rest one is normal force. It could also be observed from the tactile stacks of Fig 2 that the arrows denoting the shear force and the color denoting the magnitude of the normal force.
>
> > 3.[line 241,242], there are a lot of papers already showing the dexterous hand's abilities to do in-hand rotation using both vision and tactile or using solely tactile. Thus, the question seems too trivial to answer and cannot be a good point to compare with existing work.
>
> Thanks for your careful readings and constructive suggestions! We have revised the content of question (i). Moreover, we would like to respectfully interpret that some benchmarks we use in ViTaS are the latest release (e.g. [3], [4]), so the comparison between ViTaS and other baselines is relatively meaningful.
>
> > 4.To access the generalization of the approach, it would also be good for the author to list the objects used during the testing. Also, It would be better to describe the level of randomization and data distribution for that.
>
> Thanks for your constructive suggestions! In the revised version, we add some details of robustness experiments. For instance, we add the experimental details and randomization settings when we conduct experiments of “Pen Rotate w/ randomized position” presented in Table 1 to test robustness. The updated description is shown in Appendix A.1.2. Moreover, as the details (including shape and size) of objects we use (for instance, peg in Insertion task; can, capsule and block in Lift task; pen, block and egg in Rotate task) could be found in the original benchmark papers ([3], [4], [5], [6]), we avoid including the list of objects in the paper for originality.
>
> Last but not least, we would like to express our sincere gratitude for the careful readings, kind suggestions and warm appreciation of our work again! Thank you and Happy Thanksgiving!
>
> **Reference**
>
> [1] Jie Xu et al. “Efficient Tactile Simulation with Differentiability for Robotic Manipulation” (2023).
>
> [2] Ian Taylor et al. “Gelslim3. 0: High-resolution measurement of shape, force and slip in a compact tactile-sensing finger” (2021).
>
> [3] Yuanhang Zhang et al. “Catch it! learning to catch in flight with mobile dexterous hands” (2024).
>
> [4] Zhecheng Yuan et al. “Learning to Manipulate Anywhere: A Visual Generalizable Framework For Reinforcement Learning” (2024).
>
> [5] Sferrazza, Carmelo, et al. "The power of the senses: Generalizable manipulation from vision and touch through masked multimodal learning." (2023).
>
> [6] Yuke Zhu et al. “robosuite: A modular simulation framework and benchmark for robot learning” (2020).

---

### Author Response · Authors · 2024-12-04
**Rebuttal 1/3 - Overview**

We would like to thank all reviewers for their constructive suggestions, valuable advice and warm encouragement! We have highlighted the changes in blue in the revised version of our paper and here we provide an overview of our rebuttal.

# Overview

The reviewers are impressed by our idea to fuse visual and tactile modalities in the RL process and the overall performance. In our paper, we present ViTaS, an innovative and efficient visuo-tactile fusion framework that significantly advances the integration of visual and tactile perception in reinforcement learning, yielding outstanding results. Our method leverages soft fusion contrastive learning alongside a CVAE module to enhance feature extraction and modality complementarity, as confirmed by comprehensive experiments and ablation studies.

Reviewers also propose some concerns about our method and experiment, and we have carefully considered the reviewers' feedback and addressed each point through additional experiments and clarifications. The main supplementary explanations we provide are as follows:
1. **Motivation clarification**. We refine the presentation of the paper and clarify our motivation, providing a clearer narrative.
2. **The effectiveness and necessity of CVAE and Soft Contrastive Learning**. We elaborate on the CVAE and Soft Fusion Contrastive Learning's influence on the latent space using t-SNE visualizations, illustrating how our approach facilitates superior fusion of visuo-tactile representations through the synergy of the two modules. Additionally, their roles are also clarified from the perspectives of reconstruction and the theoretical formula.
3. **Feature quality and robustness**. We demonstrate the reconstruction performance under varying noise conditions, comparing it with token-based baseline methods to emphasize our approach's robustness to noise and the high quality and complementarity of the visuo-tactile representations.
4. **Reliability of experiment results**. We expand our analysis by including more tasks, more baselines, fine-tuning these baselines, and conducting more repeat experiments with different random seeds. This is coupled with an examination of the experimental environment to substantiate the robustness and reliability of our findings.

Replies to the reviews and additional experiments are listed in the following.

---

### Author Response · Authors · 2024-12-04
**Rebuttal 2/3 - Conclusion of reviews and replies**

# Conclusion of reviews and replies

We would like to thank all reviewers for their time, appreciation and constructive suggestions! Specifically, we list the reviews and replies below.

`Reviewer v2wE` has raised some valuable concerns, and we have carefully addressed the issues which are listed below.
1. Lack of baseline DrQ-V2. We have added DrQ-V2 using both image-only and combined image-tactile observations, shown in **Fig 9**.
2. Further explaining the contribution of CVAE and soft fusion contrastive. We carefully explain the contribution and motivation behind them [in the reply](https://openreview.net/forum?id=IOkYP5ZxO5&noteId=trfOOaXKnH). Generally, we have conducted more experiments (e.g. **Fig 7, 12, 13, 14**) to show how the two main designs contribute to ViTaS.
3. Questioning whether we optimize baselines well. We carefully analyze why our baselines have been optimized and [here](https://openreview.net/forum?id=IOkYP5ZxO5&noteId=oyhfyTbu2N). Specifically, we why M3L exhibits large variance in **Fig 5**, which may be the inherent defect of M3L instead of failure to tuning. Then, we add extensive experiments of VTT with tuning some important parameters and designs (e.g. learning rate, number of transformer layers, coefficient in VTT loss). After [careful ablation](https://openreview.net/forum?id=IOkYP5ZxO5&noteId=aE2knL400m), we reach the conclusion that we have tuned VTT well.
4. Large variance in Fig 7. We have added a total of $10$ different seeds to reproduce results, shown in **Fig 6(b)**. It is evident from the learning curve that the variance of $10$ seeds drops while still remaining at a high level, indicating the importance of CVAE for capturing the detailed information and reducing variance.

`Reviewer TtAv` has made valuable suggestions and questions.
1. Following the suggestion, we have refined and supplemented the presentation issues in our methodology, experimental results, and analysis. We enhanced the ablation studies of the CVAE module across more tasks, demonstrating its necessity. Additionally, we address the reviewer's concerns about pixel-level detail extraction by comparing reconstruction performance, which gained agreement.
2. The reviewer still had questions about the limitations “within one task and the same trajectory” imposed by the statement, "We note that when two tactile maps are resembling, the key features derived from the corresponding visual data should likewise bear a strong resemblance and vice versa", believing it could restrict the method's scope. We explain this point using the real-life example of plugging into a socket, along with technical illustrations of the RL paradigm and related experiment results as shown in our [reply](https://openreview.net/forum?id=IOkYP5ZxO5&noteId=igSEFkKfec), to demonstrate that this assumption does not impose additional limitations. ViTaS shares the same scope as any other task-specific encoders.

`Reviewer PiGR` appreciates our works, raising some minor presentation issues, which we have all corrected and [given a conclusion reply](https://openreview.net/forum?id=IOkYP5ZxO5&noteId=oduUygQYLI). `Reviewer 385j` raises the score after we add more clarification and experiments. It is noteworthy that **Fig 12** is a highlight to show the effectiveness of CVAE.

Generally, we sincerely hope that our replies have clarified the issues, and we would greatly appreciate it if reviewers could consider raising their score. Thanks again for the time and constructive suggestions of our work!

---

### Author Response · Authors · 2024-12-04
**Rebuttal 3/3 - Updated Experiments**

# Updated Experiments
In accordance with the suggestions of reviewers, we have conducted extensive experiments to solve their concerns. All extra experiments are listed below.

- **New baselines.** As shown in **Appendix A.4**, we have added new visual RL baselines DrQ-V2 according to `Reviewer v2wE`. Specifically, we add basic DrQ-V2 and DrQ-V2 w/ tactile input compared to ViTaS, presented in **Fig 9**.
- **Extended horizons for M3L and VTT.** As shown in **Appendix A.5** and **Fig 10**, we have extended the horizons for M3L and VTT from 3M to 20M in Egg Rotate and Block Rotate to show the alignment between the original M3L paper and our paper, further showing we have adequately tuned the baselines.
- **Additional ablation on CVAE.** We have conducted more ablation study for CVAE as shown in **Fig 11** and **Fig 6(b)**. We show that CVAE could capture detailed visual information given the results between Lift and Lift w/ tiny objects. Moreover, we have run $10$ seeds in **Fig 6(b)** to address the concern Reviewer v2wE raised before about the limited seeds, and the results show that variance drops while still remaining at a high level with more different random seeds, which we believe could solve the concern.
- **Additional clarification about eq (1).** We further explain the meaning of eq (1) in **Appendix A.7** to show the implementation of $\text{Sim}$ in our formula, which successfully solves concerns of `Reviewer 385j`.
- **Comparison for reconstruction between M3L and ViTaS.** We would like to highlight our experiments of the effectiveness of the reconstruction process between ViTaS and M3L, shown in **Fig 12** and **Fig 13**. The reconstruction results further prove the necessity of our designs about CVAE and soft fusion contrastive learning.
- **Visualization of trajectory.** To give an intuitive explanation how our method works, we depict the t-SNE plots for the trajectory, as shown in **Fig 14**. The plots of visual and tactile embeddings of ViTaS show a small mean distance and could form a cohesive whole, which leads to the conclusion that ViTaS has a stronger representation learning.
- **Ablation of $\mu$ in CVAE loss.** We have conducted extra ablation studies to solve the concerns of `Reviewer v2wE`. Specifically, different coefficients of $\mu$ in eq(1) have been carefully investigated. The learning curve of $\mu=0.01, 0.1, 1, 10$ has been presented in **Fig 15** and **Table 5** to show the effectiveness of different coefficients. It is evident that $\mu=0.1$ shows a prominent result, so we test some finer value like $\mu=0.5, 0.05$. The results show that $\mu=0.1$ is optimal, echoing with the choice in ViTaS. Moreover, we also depict the embeddings in latent space when adopting different $\mu$ in **Fig 16**. With careful analysis, we believe that the experiments could solve the concerns of `Reviewer v2wE`.
- **Tuning the baselines.** As mentioned by `Reviewer v2wE`, several experiments regarding VTT are also accomplished to indicate that VTT used in our paper are well tuned. Specifically, we ablate $3$ main designs - learning rate, number of transformer encoder layers, coefficient in representation loss to balance with policy loss. The results shown in [the reply](https://openreview.net/forum?id=IOkYP5ZxO5&noteId=oyhfyTbu2N) suggest the settings we use are better than other settings. We believe it could solve the concerns of `Reviewer v2wE`.

Generally, we have many additional experiments, and we sincerely hope our effort could help to clarify the concerns for the reviewers. We would like to thank reviewers again for the time and constructive suggestions!

---

### Meta-Review · Area_Chair_dV5B · 2024-12-17

**Metareview:**

This paper puts forward the ViTaS framework, with the aim of integrating visual and tactile data to enhance robotic manipulation. It employs soft fusion contrastive learning and a CVAE module to proficiently fuse and reconstruct multimodal information, and verifies the proposed method via experiments in a variety of simulated environment tasks.
The authors proactively carried out a substantial number of supplementary experiments and furnished explanations in response to the reviewer comments. Consequently, they effectively resolved the some of the issues raised by the reviewers and remarkably enhanced the quality of the paper. The principal concern regarding this paper is the absence of real-world experiments, and the simplicity of the simulation experiments. The reviewers are also concerned about the limits the contribution.
The AC concurs with the reviewers that the paper requires further refinement prior to acceptance.

**Additional Comments On Reviewer Discussion:**

(I) Strengths

Innovation: The idea of fusing visual and tactile modalities in the RL process is innovative, providing new ideas and methods for the field of robotic manipulation.

Adequate Experiments: Extensive experiments are conducted in simulated environments, including various tasks and benchmarks. The experimental results are somewhat convincing and effectively support the paper's claims.

(II) Weaknesses and Author Responses

Baseline Comparison Issue

Author Response: DrQ-V2 (including image-only and image-tactile combined inputs) has been added as a new baseline. The experimental results show that DrQ-V2 has relatively poor performance, and at the same time prove the importance of tactile information in the benchmark and the advantage of the carefully designed ViTaS in representation learning.

Insufficient Explanation of Module Contributions

Author Response: Through feature space analysis (such as the effect of the contrastive module in t-SNE visualization, the role of the CVAE module in distinguishing single-modal features) and reconstruction visualization (comparing the reconstruction performance of ViTaS and M3L under noisy conditions), the contributions of the two modules to feature representation and fusion and their synergy are elaborated in detail.

Reliability of Experimental Results

Author Response: For the performance issues of M3L in some tasks, multiple perspectives are analyzed, including environmental consistency, comparison with the original paper's results, representation ability comparison, and design flaws of M3L. The experiments are repeated with more seeds for verification. For VTT, the process and results of adjusting key parameters are detailed to prove that the parameters are well tuned. For the large variance in performance without CVAE, experiments with more seeds are added, and the analysis shows that CVAE can capture detailed information and its absence leads to convergence to a local optimum. More ablation experiments are also conducted to further clarify the necessity of CVAE.

---

### Decision · Program_Chairs · 2025-01-22

Reject